# Activation mechanism of a small prototypic Rec-GGDEF diguanylate cyclase

Raphael D. Teixeira [ID] [1], Fabian Holzschuh[1] & Tilman Schirmer [ID] [1✉]

Diguanylate cyclases synthesising the bacterial second messenger c-di-GMP are found to be regulated by a variety of sensory input domains that control the activity of their catalytical GGDEF domain, but how activation proceeds mechanistically is, apart from a few examples, still largely unknown. As part of two-component systems, they are activated by cognate histidine kinases that phosphorylate their Rec input domains. DgcR from *Leptospira biflexa* is a constitutively dimeric prototype of this class of diguanylate cyclases. Full-length crystal structures reveal that $BeF_3^-$ pseudo-phosphorylation induces a relative rotation of two rigid halves in the Rec domain. This is coupled to a reorganisation of the dimeric structure with concomitant switching of the coiled-coil linker to an alternative heptad register. Finally, the activated register allows the two substrate-loaded GGDEF domains, which are linked to the end of the coiled-coil via a localised hinge, to move into a catalytically competent dimeric arrangement. Bioinformatic analyses suggest that the binary register switch mechanism is utilised by many diguanylate cyclases with N-terminal coiled-coil linkers.

[1] Structural Biology, Biozentrum, University of Basel, Basel, Switzerland. ✉email: tilman.schirmer@unibas.ch

C-di-GMP is a near-ubiquitous bacterial second messenger responsible for the coordination of a variety of cellular processes and behaviour, including motility, biofilm formation, virulence and cell cycle progression[1]. Intracellular levels of c-di-GMP are regulated by the opposing actions of diguanylate cyclases (DGCs), which contain a GGDEF domain and synthetise c-di-GMP, and phosphodiesterases (PDEs), responsible for the degradation of this second messenger via an EAL or HD-GYP domain[2]. Not unfrequently, dozens of these enzymes can be encoded by one single genome with each of the proteins containing distinct sensory input domains that can sense/receive diverse signals like $O_2$, light and metals[3–5]. This allows bacteria to detect intracellular and environmental cues and respond promptly by adjusting c-di-GMP levels, which will then be detected by specific receptors. Common input domains are GAF[6] and PAS[7] that can recognise a variety of molecules, and response regulator receiver domains (Rec)[8], which as part of two-component systems are phosphorylated by cognate histidine kinases (HKs)[9].

DGCs catalyse the condensation of two molecules of GTP to yield the twofold symmetric c-di-GMP product. This requires the juxtaposition of two GTP molecules bound to DGC domains in appropriate twofold related arrangement to form a catalytically competent GGDEF dimer that enables nucleophilic attack of the deprotonated O3' hydroxyl onto the phosphorous of the other GTP molecule[10]. The first characterised full-length DGCs were PleD and WspR of Rec-Rec-GGDEF and Rec-GGDEF domain organisation, respectively, which were also studied in the beryllofluoride ($BeF_3^-$) modified form known to mimic phosphorylation[11]. It was shown that, upon this modification, PleD shifts from a monomer to a catalytically active dimer[12], whereas the behaviour of WspR was more complex in that it enhanced tetramer formation[13]. Later, structural and biochemical analyses on DGCs with other input domains revealed that these enzymes can exist also as constitutive dimers. Zinc binds to the CZB domain of DgcZ and prevents productive encounter of the GGDEF domains by restraining domain mobility[5]. DosC has a globin domain with bound haem to sense oxygen[3], and lastly, the bacteriophytochrome PadC senses red light through its PAS-GAF-PHY domains to activate the GGDEF domain[14].

Almost invariably, input and catalytic domains of DGCs are connected by a dimeric coiled-coil that can vary in length. We proposed earlier that the constituting helices could change their crossing angle and/or azimuthal orientation to allow or prevent productive encounter of the two GGDEF domains (chopstick model[10]). This mechanism would be a generalisation of the scissors with fixed pivot blades model ascribed to histidine kinases signalling[15]. To test for the mechanism, we selected a prototypical minimal DGC with known input signal. LEPBI_RS18680 from *Leptospira biflexa*, hereafter called DgcR (diguanylate cyclase controlled by Rec), is a Rec-GGDEF protein with a short domain linker (Fig. 1a), thus making this enzyme attractive for studying its conformational states by crystallography.

*Leptospira* is a bacterial genus composed of more than 30 species, among them some pathogenic representatives responsible for causing leptospirosis, a worldwide zoonosis that affects more than one million people and accounts for 60,000 deaths per year[16]. *Leptospira biflexa* is a saprophytic species used as a model to study *Leptospira* biology[17]. It contains an additional extra-chromosomal element of 74 kb (p74) that codes for 56 proteins including DgcR. DgcR was also chosen, because it shares the same domain organisation and linker length as Rrp1 from *Borrelia burgdorferi*, the pathogen responsible for causing the Lyme disease. Rrp1 is the only DGC encoded by *B. burgdorferi* genome and is essential for bacterial survival in the tick host[18].

Here, we show by biophysical and crystallographic analyses that DgcR is a constitutive dimer that changes coiled-coil geometry and domain arrangement upon pseudo-phosphorylation. The chopstick model is generally confirmed, but, upon activation, DgcR shows an unexpected translational register shift. Bioinformatic analyses suggest that the observed activation mechanism is most likely operational in most diguanylate cyclases of Rec-GGDEF organisation, but also in some other DGCs.

## Results and discussion

**DgcR is a constitutive dimer that is activated by domain rearrangements.** To reveal the structural changes accompanying the activation of Rec-GGDEF DGCs we determined the full-length crystal structures of DgcR in native and pseudo-phosphorylated ($BeF_3^-$ modified) state. A DgcR variant (R206A/D209A, abbreviated DgcR_AxxA) that had the putative allosteric inhibition site (Fig. 1a) mutated was used to avoid locking the enzyme in a product-inhibited conformation. Crystallisation was performed in presence of 3'-deoxy-GTP (3'dGTP), which is a non-competent substrate analogue due to the absence of the 3'-hydroxyl group.

The structure of native DgcR_AxxA (called DgcR') was solved by molecular replacement to 2.2 Å resolution (Supplementary Table 1). There is one dimer in the asymmetric unit with the protomers held together by extensive isologous contacts between the Rec domains involving their α4-β5-α5 face (Fig. 1b). Notably, the electron density of the Rec domains is considerably weaker than that of the GGDEF domains, indicating a larger mobility. The Rec domain shows the canonical $(\beta\alpha)_5$ fold (rmsd of 1.5 Å for 116 Cα atoms with respect to PhoP, 2PKX), but with the C-terminal α5-helix considerably extended and forming together with its symmetry mate a coiled-coil, leading to the GGDEF domains. A $Mg^{2+}$ ion is bound to the acidic pocket formed by E12, D13, and the phosphorylatable D56.

The structure of the GGDEF domain is very similar to others in the PDB database (rmsd of 1.4 Å for 157 Cα atoms with respect to PleD, 2V0N) and shows the canonical (β1-α1-α2-β2-β3-α3-β4-α4-β5) topology of nucleotidyl cyclases of group III[19] with an N-terminal extension that starts with a characteristic wide turn showing a DxLT motif followed by helix α0 that leads to β1 (Fig. 1b, see also ref. [10]). The GG(D/E)EF motif is located at the turn of the β2-β3 hairpin. Again as observed in other structures[12], the guanine base of the substrate analogue is bound to a pocket between α1 and α2 and forms H-bonds with N182 and D191, whereas the two terminal phosphates are H-bonded to main-chain amides of the short loop between β1 and α1. Additionally, the γ-phosphate forms ionic interactions with K289 and R293. Two magnesium ions are bound to the usual positions being complexed to the β- and γ-phosphates and the side-chain carboxylates of D174, E217, and E218.

The GGDEF domains do not obey the twofold symmetry of the Rec domains, but form a relative angle of about 90°. Thus, the two active sites with the bound GTP analogues do not face each other rendering this constellation clearly non-productive. Though the constellation may be determined to some extent by crystal packing, it demonstrates considerable inter-domain flexibility. Comparison of the main-chain torsion angles reveals that the relative rotation can be traced back to a 169° change in a single torsion, namely around the Cα–C bond of residue 136 (ψ136, Fig. 2a). Thus, the hinge locates to the C-terminal end of Rec α5, with the following residue I137 being packed against the Y149 from the beginning of the GGDEF α0' helix in both chains (Fig. 2b, c). As noted before[10], the conserved residue N146 (Supplementary Fig. 6) is capping both α5 and α0', but only in the A-chain.

The structure of activated DgcR_AxxA (called DgcR'*) obtained by $BeF_3^-$ modification was solved by molecular replacement to 2.8 Å resolution (Fig. 1c) (Supplementary Table 1). There are two dimers in the asymmetric unit that show virtually

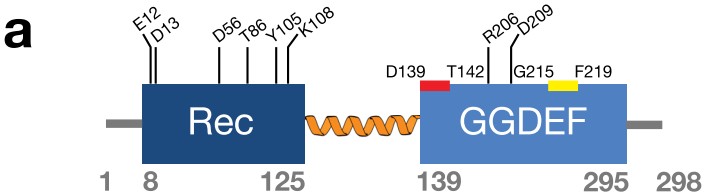

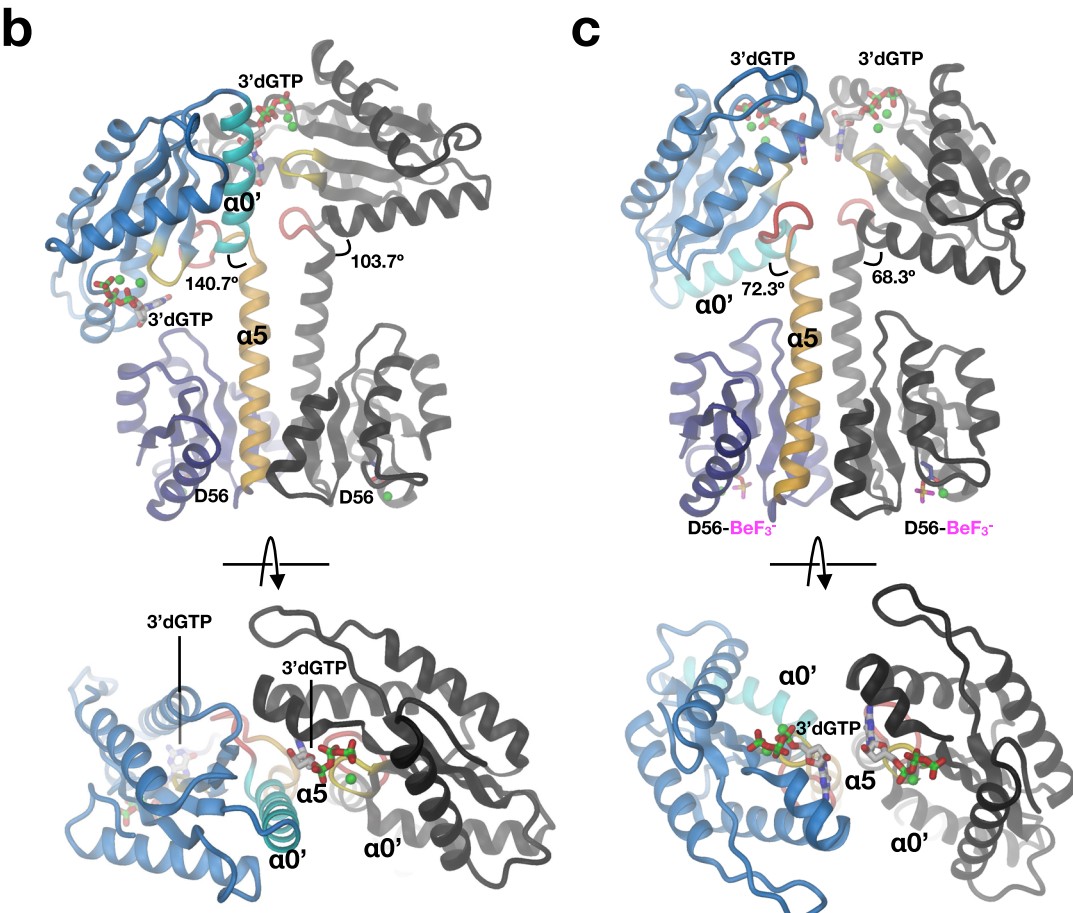

**Fig. 1 Native and activated DgcR dimers adopt distinct domain arrangements. a** DgcR domain organisation with important features and residues highlighted. The Rec and the GGDEF domains are linked by the extension (orange) of the C-terminal Rec helix. Red and yellow bars indicate the DxLT and GGDEF motif, respectively. **b, c** Side and top views of DgcR' (**c**) and DgcR'* (**d**) dimers. Within one protomer, domains and important elements are highlighted by colour. The C-terminal Rec helix (α5) is coloured in gold, the wide turn in red, the N-terminal GGDEF helix (α0') in cyan and the characteristic β-hairpin (with GGEEF sequence) in yellow. The BeF$_3^-$ modified aspartates of the Rec domains, the bound Mg$^{++}$ ions (green), and the 3'dGTP substrate analogues bound to the GGDEF active sites are shown in full. In both cases, the Rec domains obey twofold symmetry. The GGDEF domains are related by twofold symmetry (with the two 3'dGTP ligands opposing each other) only in DgcR'*, while in DgcR' they are related by ~90°.

the same Rec dimer structure, but slightly different α5-helix bending and GGDEF orientations (Supplementary Fig. 1). As in DgcR', the dimer is formed by isologous contacts between the α4-β5-α5 Rec faces and the extension of α5 forms a coiled-coil, but with an altered relative disposition, which will be described in detail further below. D56 is found fully modified by BeF$_3^-$ and its immediate environment is different compared to DgcR' as will be discussed in detail hereafter. The GGDEF domains are arranged symmetrical with the two bound 3'dGTP ligands facing each other, but too distant for catalysis (Fig. 1c, bottom). The GGDEF orientation relative to the Rec domain is similar as in the A-chain of DgcR'. As in the DgcR' structure, there are no direct contacts between the domains of a protomer.

Consistent with the crystal structures and the presence of the coiled-coil in both states, in solution, DgcR is a constitutive dimer as measured by Multi-Angle Light Scattering (MALS) both in the native and the activated form (Supplementary Fig. 2). Addition of substrate analogue, product or lowering salt concentration did not change the quaternary state.

**Aspartate modification induces a relative rigid-body rotation within the Rec domain.** Comparison of native and activated DgcR' (Fig. 3a) shows that, on activation, the hydroxyl group of T86 is moved towards the BeF$_3^-$ moiety to form an H-bond. The void left by this movement is claimed by Y105 that undergoes a small side-chain rotation, but does not change its rotamer.

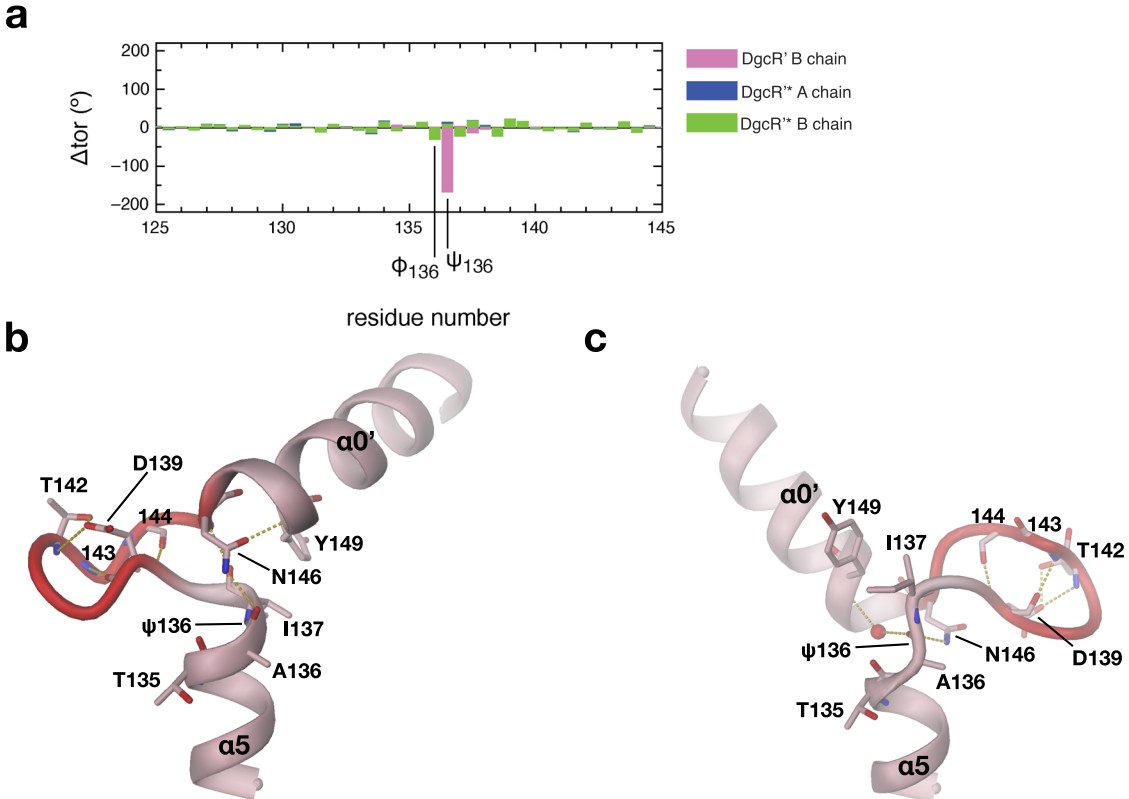

**Fig. 2 Inter-domain hinge revealed by comparison of the two DgcR' protomers. a** Difference of backbone torsion angles (Δtor) of DgcR' and DgcR'* chains relative to chain A of DgcR'. Note that the two DgcR' chains show a localised drastic difference in the main-chain torsion angle ψ136. **b, c** Detailed view of the inter-domain hinge segment of chain A (**b**) and B (**c**) of DgcR'. The wide turn with the D139xLT142 motif is highlighted in red. The 169° rotation of ψ136 drastically changes the relative angle between α5 of the Rec and α0' of the GGDEF domain.

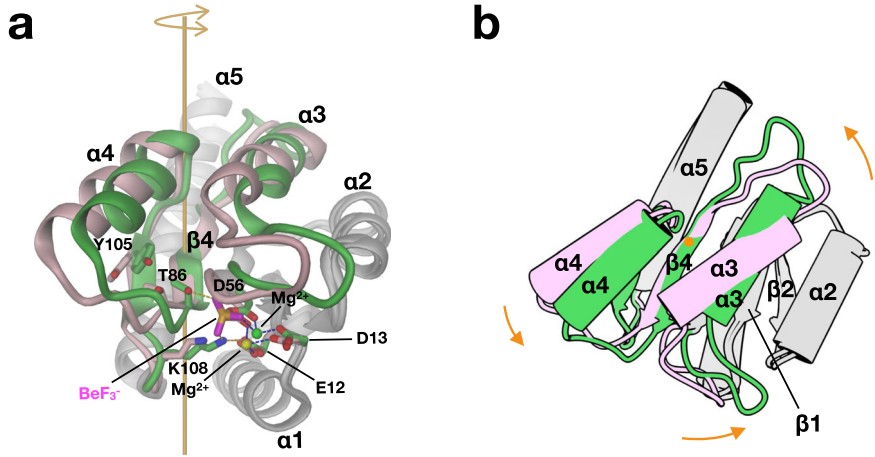

**Fig. 3 Beryllofluoride modification induces a relative 16° rotation of two Rec halves.** Rigid-body 1 composed of α3, β4, α4, β5 (shown in pink and green for DgcR' and DgcR'*, respectively) is rotated with respect to the rest (rigid-body 2, grey) as seen after super-postion of the two grey substructures. **a** Native and activated Rec domains with important residues shown in full viewed perpendicularly to the rotation axis of the relative rotation (orange). In DgcR'*, the beryllofluoride group forms an H-bond with T86 and an ionic interaction with K108 of DgcR'*. **b** Same as **a**, but in cartoon representation and viewed along rotation axis.

Furthermore, K108 forms ionic interactions with BeF₃⁻ and E12 in the activated structure (Fig. 3a). In the native state, a magnesium ion is bound loosely to E13 and D56, whereas, in the active state, it is additionally coordinated by the BeF₃⁻ moiety.

Activation of DgcR is accompanied with a change in the backbone structure as identified by a DynDom analysis[20]. The Rec domain can be divided into two parts that undergo a relative 16° rotation as shown in Fig. 3b. Thereby secondary structure elements α3 to β5 (residues 54 to 108) behave as one rigid body (rmsd = 0.83 Å/49) that rotates relative to the rest (8–53, 109–135) that superimposes with an rmsd of 1.19 Å for 67 Cα positions. Figure 3b shows that the rotation axis passes roughly perpendicular to the β-sheet through the centre of β4 (L83). Note that the phosphorylatable D56 is close to the junction between the two rigid bodies and that its Cα position only changes slightly during the transition. T86, however, with its distance of 7.5 Å

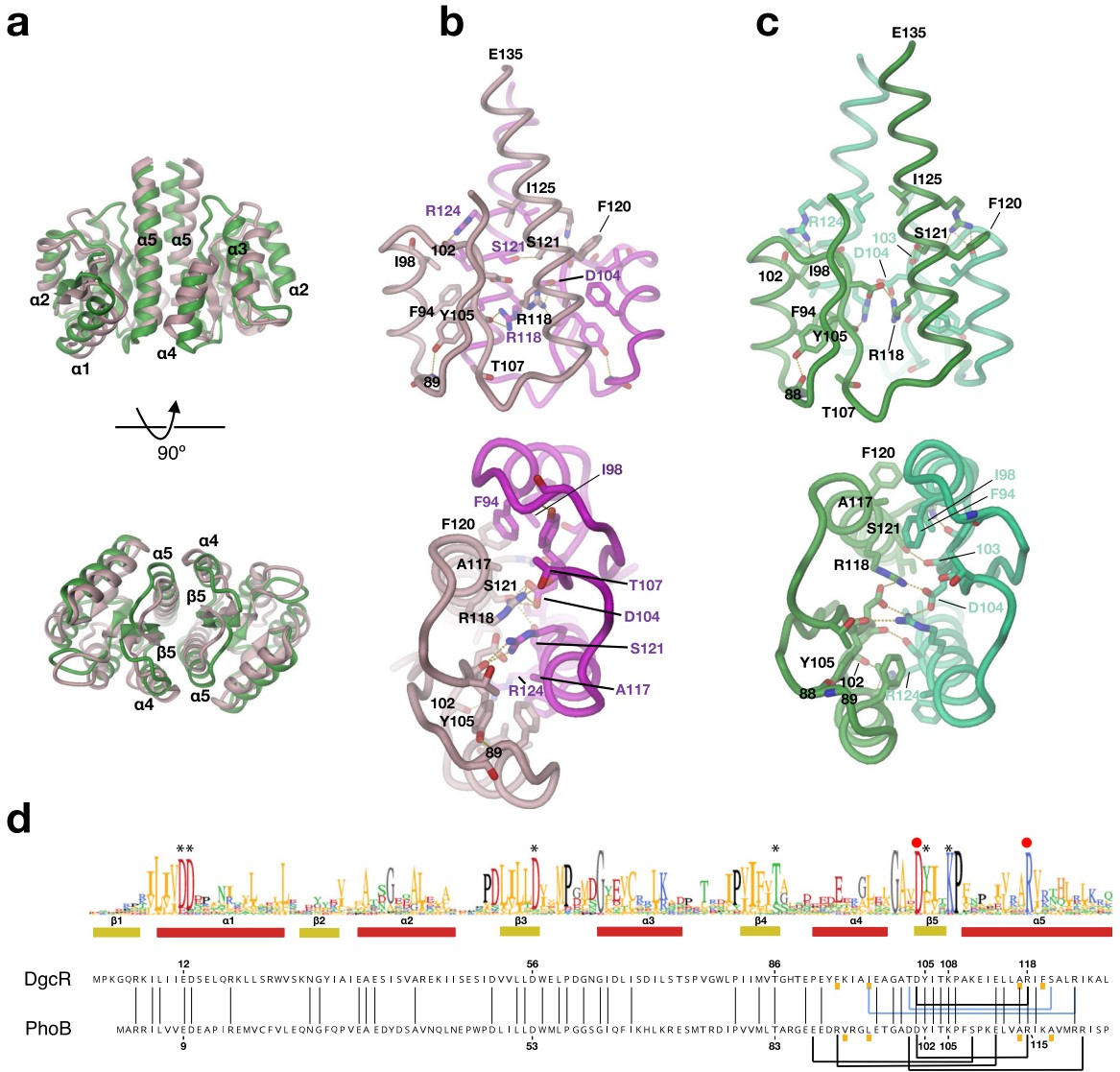

**Fig. 4 Distinct packing of Rec domains upon BeF3- modification. a** Superimposed dimers with native structure in pink and activated structure in green. **b**, **c** Rec dimer of DgcR' (**b**) and DgcR'* (**c**) with contacts between the protomers <3.2 Å in yellow. **d** Sequence alignment of Rec domains of DgcR and PhoB with secondary structure elements of DgcR and sequence logo derived from DgcR group 1 homologues (see Fig. 10). Asterisks indicate conserved residues involved in Rec domain activation. Below the individual sequences, lines connect residues that participate in inter-domain contacts (side-chain–side-chain interactions in black, interactions involving the main-chain in blue). Red dots indicate D104 and R118 of the conserved intermolecular salt-bridge.

from the rotation axis moves by 2.0 (Cα) to 3.3 Å (Oγ) and the motion is most pronounced (5.2 Å) for the N-terminus of α3 (P91) with its distance of about 15 Å from rotation axis. Thus, the rigid-body motion changes significantly the arrangement of α4 with respect to α5, which has a profound effect on the packing of the Rec domains in the dimer.

For many Rec domains, a Y-T coupling mechanism has been described, where, upon (pseudo-) phosphorylation a threonine/serine (T86 in DgcR) is dragged towards the phosphate and the conserved tyrosine/phenylalanine (Y105 in DgcR) follows suite with a rotameric change from *gauche*[+] to *trans*[12,21,22]. In DgcR, the conserved tyrosine is already in *trans* conformation before activation and the T and Y move concertedly towards the berylloflouride moiety as part of a rigid-body (α3 to β5) movement (Fig. 3).

**Rigid-body rotation induces repacking of Rec domains within the dimer.** In both states, the Rec domains form twofold symmetric dimers with the contacts mediated by isologous

interactions between the α4 - β5 - α5 surfaces (Fig. 4). However, due to the rigid-body motion within the protomer and the concomitant relative displacement of α4 and α5 (Fig. 3), the association of the α4-β5-α5 faces is different in the native and the activated state. Therefore, the two dimers superimpose rather poorly (rmsd = 3.1 Å/119 Cα positions) with the β-sheets of the protomers showing a difference in orientation of about 15° (Fig. 4a).

The native Rec dimer (Fig. 4b) with a buried surface area of 980 Å² is held together by an extended apolar contact of α5 (A117, F120) with α4 (F94, I98), an ionic interaction of D104 with R118, an H-bond between main-chain carbonyl 102 and R124 (both β5-α5 contacts). All aforementioned residues are well-defined with the exception of the R118 side-chain, which probably has several alternative conformations, but all placing the guanidinium group close to D104 and to its symmetry mate. Finally, and most relevant for the allosteric regulation of the C-terminal GGDEF effector domains, there are regular coiled-coil interactions across the symmetry axis between the C-terminal

halves of the α5 helices starting with S121. These will be discussed in the next chapter.

The activated Rec dimer (Fig. 4c) with a buried surface area of 850 Å$^2$ shows the same apolar α5-α4 cluster as the native dimer, but with the residues repacked in-line with the aforementioned relative displacement of α4 and α5 within the protomer. At the centre of the interface, D104 shows a well-defined, intermolecular salt-bridge with R118, but also with R118 from the same chain. As in the native dimer, the R124 and S121 side-chains form intermolecular H-bonds, but with other partners compared to the native interactions (main-chain carbonyls of 98 and 103, respectively).

A BLAST search revealed that, apart from Rec-GGDEF orthologs, the sequence of the DgcR Rec domain is most similar to that of OmpR-like transcription factors (Fig. 4d). These have a Rec-DNA-binding domain architecture and have been shown to dimerise via the Rec α4-β5-α5 face upon activation to allow binding of their effector domains to DNA[23]. Indeed, a structure search of the DgcR'* dimer against the PDB retrieved as top hit (rmsd = 1.5 Å/228 Cα positions) the BeF$_3^-$ activated Rec domain of PhoB (1ZES)[22]. Most of the intermolecular interactions are thereby conserved, in particular the central salt-bridge D109 - R118 (DgcR numbering), or conservatively replaced (Fig. 4d). To our knowledge, no response regulator with a DNA binding effector domain has yet been observed as a constitutive α4-β5-α5 dimer (for a review, see ref. [8]), which is probably due to their small or absent coiled-coil linkers. A special case is the Rec-Rec'-GGDEF protein PleD, where, in the activated state, the two Rec domains of each chain form a quasi twofold intra-molecular dimer involving their α4-β5-α5 faces and exhibiting OmpR like inter-domain salt-bridges[12]. Such assembled Rec-Rec' domains are then instrumental for the subsequent dimerisation of two protomers and, thus, enzyme activation. Summarising, beryllo-fluoride$^-$ modification of D56 induces a relative rigid-body motion in the Rec domain that changes the relative disposition of α4 and α5. Consequently, since both helices are part of the Rec–Rec interface, the relative arrangement of the protomers and, thus, of the two α5 helices of the dimer is changed (compare top panels of Fig. 4b, c). This change is supposed to be crucial for the allosteric regulation of the C-terminal GGDEF domains as will be discussed in the following.

**Lateral shift of C-terminal Rec helices changes their coiled-coil register**. The DgcR Rec α5-helix is longer by about three turns (10 residues) compared to that of canonical Rec domains. In the dimer, these protrusions form a twofold symmetric coiled-coil both in the native and the activated state (Fig. 1b, c), though with distinct relative arrangement. Both constellations are stabilised by isologous contacts between predominantly hydrophobic residues that obey a heptad repeat pattern (Fig. 5a, b). Thereby, I125 and L132 contribute to the contact in both structures (position a; persistent contacts), but with the side-chains interacting with their symmetry mates from opposite sides depending on the state (see, e.g., the 132–132 contact in Fig. 5a). In contrast, other residues contribute either only to the native (L128, T135) or the activated (H129, A136) constellation (positions d, e; conditional contacts).

The two contact modes represent alternative knobs-into-holes packing as best seen in the helical net diagram of Fig. 5c, suggesting a relative lateral translation of the interacting helices. Indeed, superposition of one of the helices as in Fig. 5d, e reveals a large lateral shift of about 9 Å. In other words, upon activation, the two helices do not roll over each other (which would be accompanied by a change in their azimuthal angles), but are translated with respect to each other to realise an alternative

knobs-into-holes packing. Note, that for steric reasons this shift would require dissociation and reassemble of the constituting helices. Thus, the coiled-coil behaves like a binary switch that can assume two clearly defined states, i.e., two distinct registers.

Recently, an analogous transition in the coiled-coil linker of a diguanylate cyclase has been proposed for phytochrome-regulated PadC[14]. Indeed, the C-terminal end of the coiled-coil of the dark-state enzyme is in the same register as native DgcR with, amongst others, N518 and L525 forming (conditional) contacts (see PadC in Supplementary Fig. 4). Inspection of the linker sequence and dynamic considerations prompted the authors to propose an alternative register involving the neighbouring residues N519 and A526 for the illuminated state. Indeed, mutations designed to stabilise this second register were constitutively active and the coiled-coil was found in the active register (Supplementary Fig. 4)[24]. Although the structure of light-activated wild-type PadC is not known, it is very likely that PadC and DgcR use the same binary coiled-coil switch mechanism for DGC regulation, despite unrelated input domains.

WspR, another well-studied Rec-GGDEF diguanylate cyclase, also exhibits a "slippery" axxdexx heptad repeat with e of the last repeat in position −3 with respect to the DxLT motif (Supplementary Fig. 4). Unfortunately, only product bound structures are available that reveal a non-productive, c-di-GMP cross-linked tetramer in which the coiled-coils emanating from the two Rec dimers are splayed apart at their ends[25]. Most revealing, however, a GCN4–GGDEF WspR with GCN4 interface residues in the active register (Supplementary Fig. 4) was reported to be highly active and a corresponding structure (compact dimer) was predicted for active WspR[13]. Summarising, the change in coiled-coil registration upon DgcR activation is accompanied by a substantial lateral shift of the constituting helices, which lead directly to the catalytic domains. Structural data on other DGCs are consistent with this finding.

**Small rotation around inter-domain hinge allows formation of competent GGDEF dimer**. In the activated structure, the two GGDEF domains show no mutual interactions and their precise orientation appears to be determined by crystal contacts. However, the two bound 3'dGTP ligands face each other, though their distance (>10 Å) is clearly too large for catalysis (see Fig. 1c). Having identified the CA-C main-chain bond of A136 as an inter-domain hinge (Fig. 2a), we tried, by small changes in ψ136 and adjoining main-chain dihedrals, to symmetrically move the GGDEF domains as rigid bodies into a catalytically competent arrangement (Michaelis–Menten complex). Indeed, only small torsional changes (Fig. 6a) were necessary to bring the (reconstructed) 3'-hydroxyl groups of each bound substrate in line with the scissile PA–O3A bond of the other substrate as required for catalysis (Fig. 6b, c). It should be considered, however, that the optimal arrangement of the catalytic domains depends obviously on the conformation of the bound substrates, as discussed in ref. [5]. In fact, comparison of the bound 3'dGTP ligands in DgcR' and DcrR'* shows variability in the ribose and α-phosphate orientation (Supplementary Fig. 5), probably due to the lack of strong interactions with the binding site. Here, we used the conformation as seen in DgcR'*. Scenarios with other substrate conformations were not explored, but the relative GGDEF arrangement would probably be similar considering the fixed hinges at the end of the activated Rec dimer.

The details of the Michaelis–Menten complex shown in Fig. 6c are consistent with the model proposed in ref. [10] with metal M2 coordinating the 2'-hydroxyl group and K179 hovering over the α-phosphate of the incoming substrate. There is no titratable residue close to the O3'-group. Most likely,

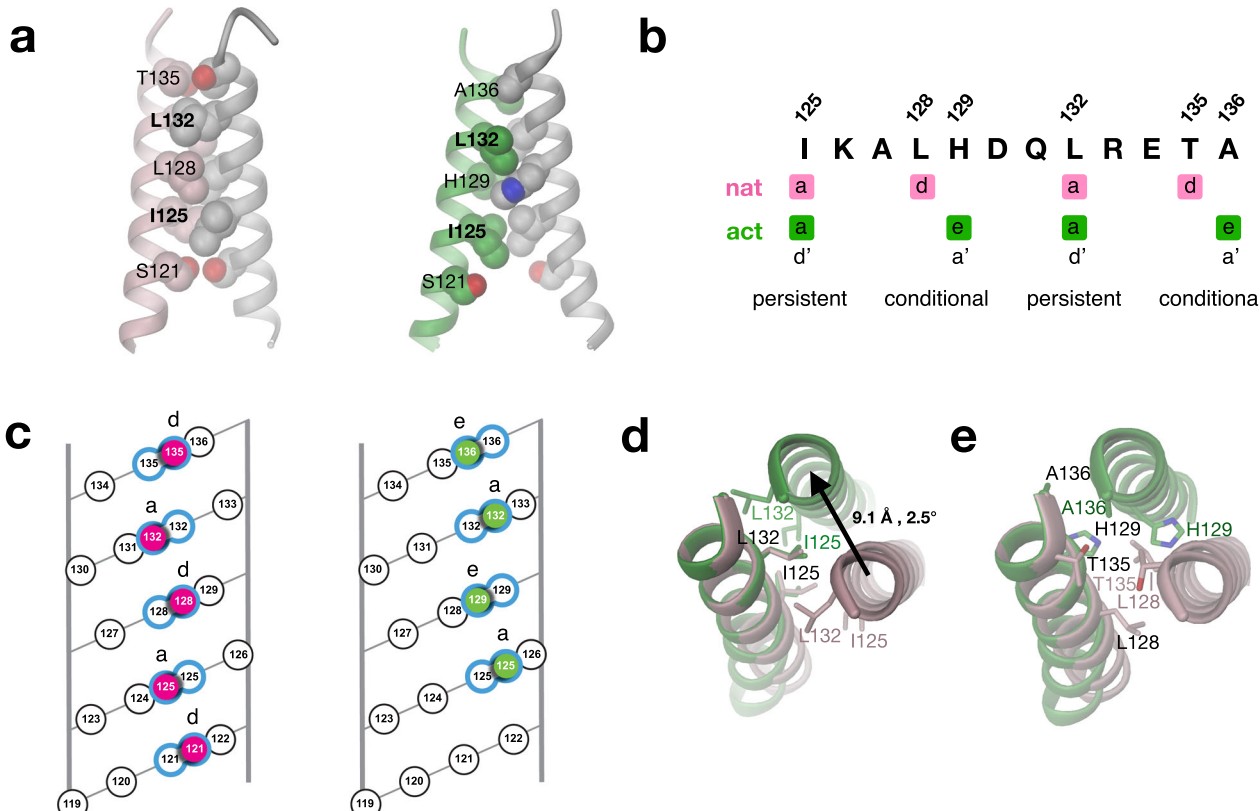

**Fig. 5 Coiled-coil linker adopts two alternative registers depending on activation state. a** Side view of parallel coiled-coil linker of DgcR' (left) and DgcR'* (right) with residues that form contacts with their symmetry equivalents in CPK representation. Only residues from the left helix are labelled. Bold labels indicate residues that are involved in both registers (persistent contacts). **b** Two alternative heptad repeat patterns (a d a d and a e a e), which are adopted by DgcR' and DgcR'*, respectively. Positions a are used in both registers (persistent contacts), whereas positions d or e are used only in the native or activated conformation (conditional contacts). Note, that the a e a e pattern can formally also be described by a d' a' d' a' pattern as indicated in the figure and used in (Gourinchas et al.)[14]. **c** Helical net representation[51] of coiled-coil interactions in DgcR' (left) and DgcR'* (right). Of the front helix, only interacting residues are shown (highlighted by colour). Interacting residues pairs are outlined in blue. **d, e** Top view of coiled-coil after superposition of left helix. DgcR' and DgcR'* are shown in pink and green, respectively. For clarity, residues forming persistent and conditional contacts are shown in the separate panels **d** and **e**, respectively.

deprotonation of the hydroxylgroup proceeds via a water molecule that could be activated by the close-by metal(s) as, e.g., in adenylate cyclases[26].

In the competent dimer, there are no clashes between the catalytic domains. Molecular dynamics simulations would be required to refine the model, but it appears that D183 and D282 may interact with Y286 and H187, respectively. All these residues are conserved in diguanylate cyclases (Supplementary Fig. 6). Indeed, in the apo-structure of the constitutively active variant of PadC ([6ET7][24]) the proposed interactions seem well possible, albeit only in one half of the asymmetric structure.

There is one more conserved residue that projects to the other subunit, namely R147 (Supplementary Fig. 6). Judged by the model, it appears possible that this arginine may interact with the guanyl Hoogsteen-edge of the opposing substrate. This would be supported by a recent study on the promiscuous (accepting GTP and ATP) DGC GacA, wherein the reason for the relaxed substrate specificity was attributed to an aspartate-serine replacement of a base-binding residue[27]. In the sub-group of promiscuous DGCs, the position homologous to R147 of DgcR is not conserved (sequence logo in Fig. 6—figure supplement 1D of ref. [27]), suggesting that the arginine is no longer important, since it cannot interact with a adenyl Hoogsteen-edge. In the same paper, the fourth residue of the GGDEF motif (equivalent to E218 in DgcR) was proposed to deprotonate the 3'-hydroxyl group of the substrate bound to the other subunit. In our model (Fig. 6c),

this residue is coordinating metal M2 and clearly not close to this substrate hydroxyl group.

**Structural coupling of Rec modification with competent dimer formation**. The knowledge of the structures of full-length DgcR both in its native and activated form and the model of the Michaelis–Menten complex allows to discuss in detail how signal perception (phosphorylation) is coupled to output activation in a prototypic response regulator with enzymatic function. In Fig. 7, this process is dissected into 5 notional steps.

(1) Starting with a symmetrized version of DgcR' (Fig. 7a), aspartate pseudo-phosphorylation induces a rigid-body motion within each Rec domain (Fig. 7b, tertiary change). With an unchanged coiled-coil packing, the intermolecular α4–α5 contacts would break up. (2) This is counteracted by a repacking of the two Rec domains (Fig. 7c, quaternary change. (3) The clashing of the C-terminal ends of the coiled-coil is relieved by slight outward bending of the helices (Fig. 7d). Obviously, these first three steps, which describe the transition of the native to the activated Rec stalk, will be tightly coupled.

The following steps invoke no direct Rec-GGDEF communication, but only an unrestricted rotation of the GGDEF domains around the inter-domain hinges. With the Rec stalk in its activated constellation, the hinges are positioned such that the GGDEF domains can attain (4) a constellation as in DgcR'*

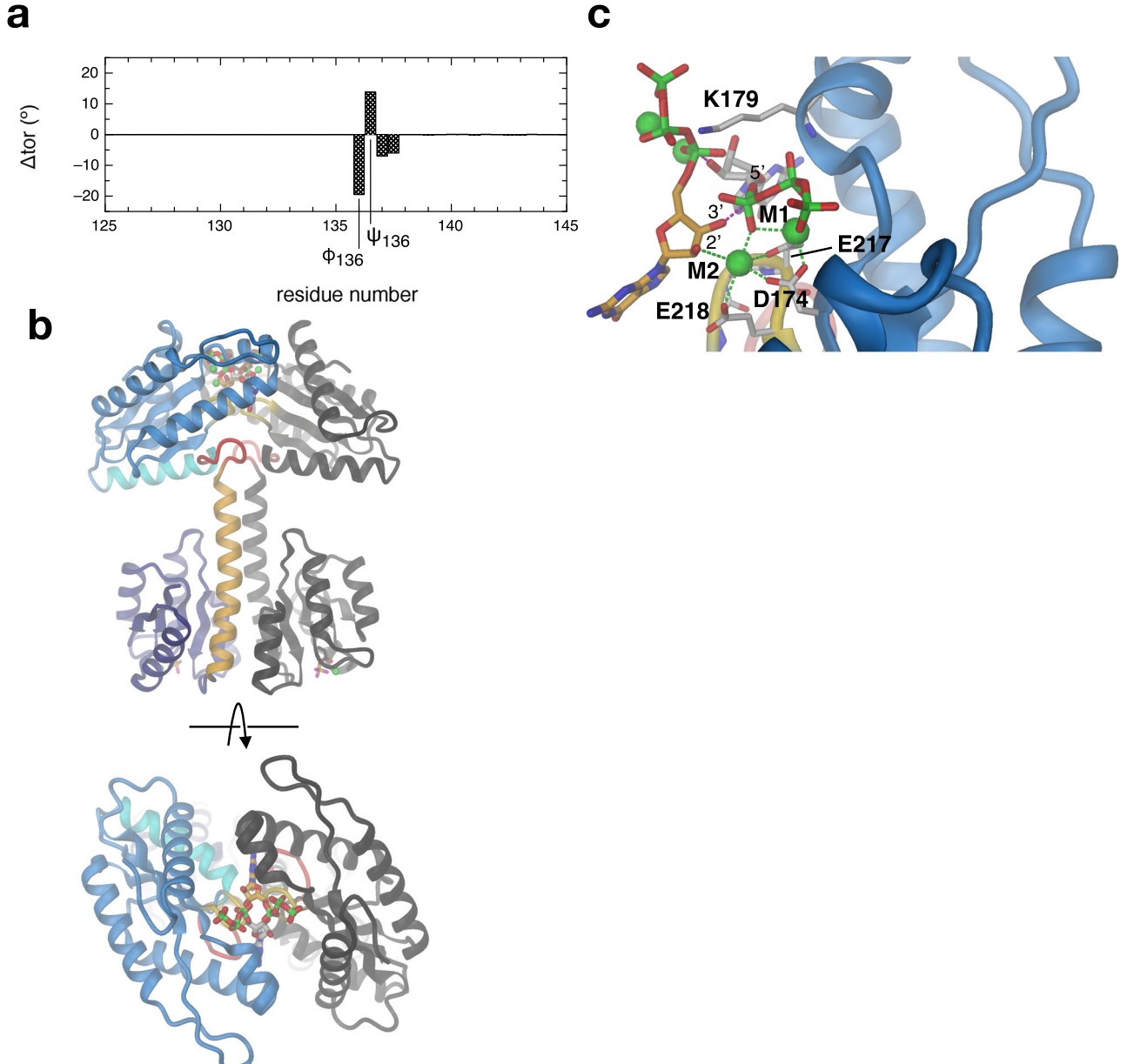

**Fig. 6 The activated Rec dimer allows formation of a catalytically competent GGDEF dimer. a** Changes in main-chain dihedral angles ($\Delta$tor) applied manually to DgcR′* to move the two GGDEF domains into catalytically competent arrangement. **b** Model of competent DgcR (two orthogonal views) generated as described in **a**. **c** Detailed view of the competent juxtaposition of the two GGDEF bound GTP substrate molecules. The carbon atoms of the two GTP molecules are coloured in orange and grey, respectively. The O3′ hydroxyl of each ligand is poised for nucleophilic attack on the α-phosphorous (PA) of the other ligand being roughly inline with the scissile PA–O3A bond.

(Fig. 7e) and, finally, assemble to form (5) the catalytically competent constellation Michaelis–Menten complex (Fig. 7f). An animation of the entire structural transition from native to competent DgcR is shown in Supplementary Movies 1 and 2. Noteworthy, the two GGDEF domains most likely won't be able to attain the productive arrangement when the Rec dimer is in the native form, since the distance between the inter-domain hinges is changed considerably upon activation/deactivation (Fig. 6b).

The aspect of conformational sampling and its dependence on the coiled-coil register and the dynamics of the entire enzyme has been discussed before for PadC[24]. Whether the asymmetric GGDEF dimer obtained for mutated PadC is of functional importance needs further studies. Although such a state would probably be compatible with our model, it is not mandatory for the proposed mechanism in which the two phosphodiester bonds

could be formed quasi-simultaneously. Furthermore, we suggest that the competent GGDEF dimer would assemble autonomously due to electrostatic and steric complementarity, in particular in presence of the substrates that interact with K179 and M2 of the opposing domain (Fig. 6c), thus not requiring any direct interaction between input and output domains.

**Allosteric inhibition by product mediated domain cross-linking**. Allosteric product inhibition by c-di-GMP is a well-known feature of many DGCs[10,12,28,29]. Hereby, dimeric c-di-GMP mutually cross-links a RxxD motif (primary I-site, $I_P$) on one GGDEF domain with a secondary I-site ($I_s$) on the other GGDEF domain and vice-versa. The crystal structure of DgcR obtained in presence of c-di-GMP (DgcR_inh) was determined to 3.3 Å by molecular replacement and is shown in Fig. 8a

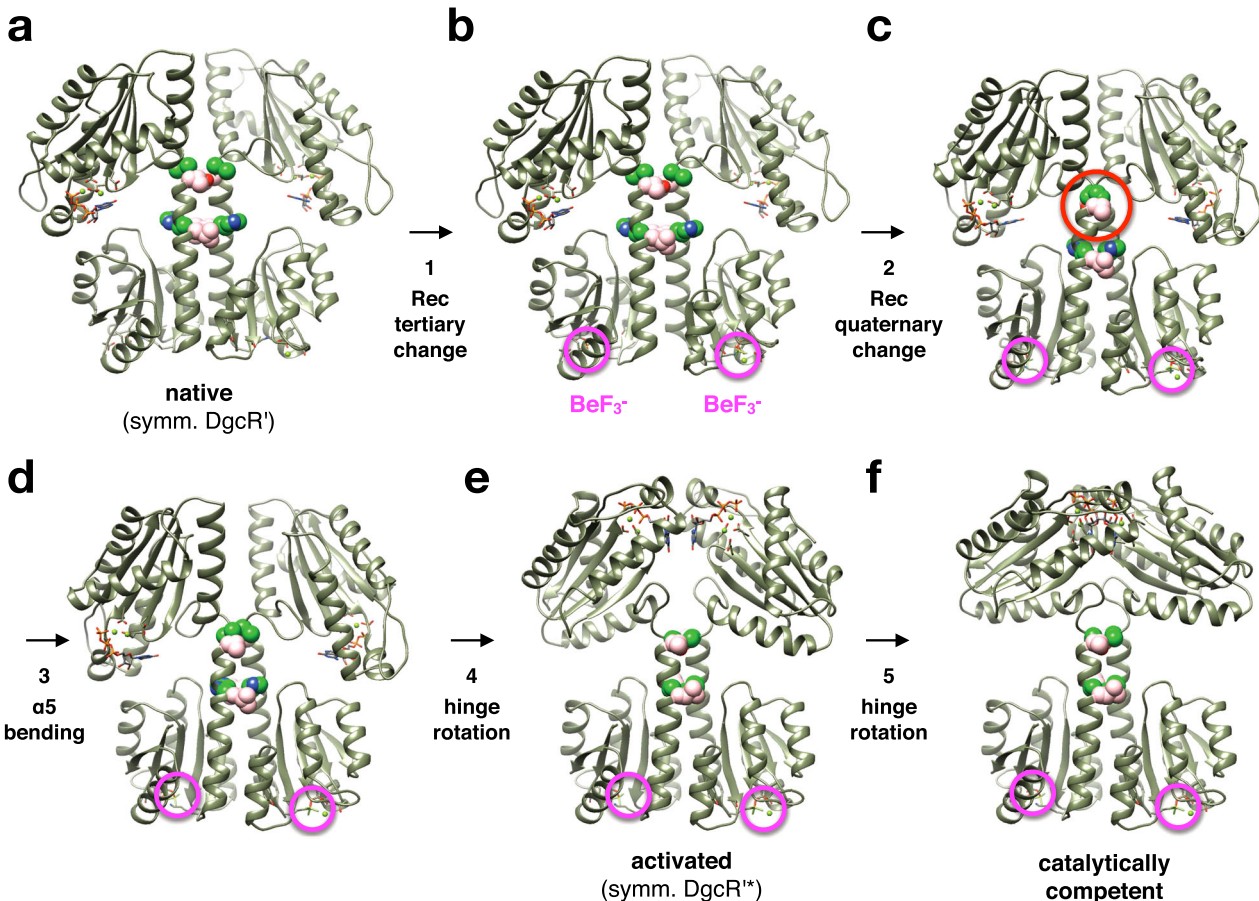

**Fig. 7 Structural transitions in DgcR upon activation.** Rec pseudo-phosphorylation induces steps 1 to 3, which are followed by GGDEF hinge motions of steps 4 and 5 to attain the catalytically competent state. See also Supplementary Movies 1 and 2. The structures are represented as in Fig. 1b, c, but with the residues of the conditional coiled-coil contacts shown as CPK models (residues in *d* and *e* position are shown is pink and green). The beryllofluoride moieties of the dimer are highlighted by magenta circles. **a** DgcR′, symmetrized version with both GGDEF domains in B-chain orientation (cf. with Fig. 1b). **b** As in **a**, but with berylloifluoride-induced tertiary change applied to Rec rigid_body 1 (see Fig. 3). **c** As in **b**, but with quaternary change applied to Rec domains. Note the clash between the C-terminal ends of the coiled-coil (red circle). **d** As in **c**, but with Rec dimer as found in symmetrized version of DgcR′*. **e** Symmetrized version of DgcR′* (cf. with Fig. 1c). **f** Model of catalytically competent DgcR as in Fig. 6b.

(Supplementary Table 1). There are three symmetric dimers in the asymmetric unit. Each dimer shows a Rec stalk in native conformation and the two GGDEF domains have their active sites facing outwards. Dimeric c-di-GMP is cross-linking the domains by interacting with the R206xxD209 motif of one subunit and R163* from α0′ of the other subunit (Fig. 8b). Owing to symmetry, there are two isologous cross-links within the DGC dimer. Comparison with the PleD/c-di-GMP complex[12] (Fig. 8c) shows very similar binding, but with PleD providing an additional arginine (R390) to the $I_p$-site. The DgcR equivalent (R237) is too distant to interact, but this may happen with the Rec stalk in the activated conformation. Arginines 163* (DgcR) and R313* (PleD) fulfil the same role in c-di-GMP binding, but are not homologous on the sequence level. Indeed, it has been noted earlier that among GGDEF sequences (Paul et al. 2007) arginines are enriched at either position. The unique c-di-GMP stabilised GGDEF arrangement that differs drastically from DgcR′ (Fig. 1b) again demonstrates the large flexibility provided by the inter-domain hinge.

**Kinetic analysis of DgcR activity reveals delay in non-competitive feed-back inhibition.** The effect of activation and $I_p$-site mutation on DgcR catalysed c-di-GMP production was studied by a real-time nucleotide quantification assay (online ion-

exchange chromatography, oIEC, Agustoni et al., in preparation, see Methods and example chromatograms in Suppl. Fig. 7). Figure 9b, c shows that product formation catalysed by native DgcR gradually decreases early-on (despite a large excess of substrate), indicative of non-competitive product inhibition. Indeed, the progress curve was found consistent with the respective classical model with a low $k_{cat}$ of about 0.01 s$^{-1}$ and a relatively large $K_i$ of about 30 µM (Supplementary Table 2).

A very different behaviour was observed for the activated enzyme (DgcR*) that produced very quickly (<75 s) a substantial amount of product (Fig. 9b, c). Figure 9a shows the initial burst as resolved by early time-point measurements using conventional IEC with EDTA to stop the reaction. The burst phase was followed by a phase of very small, virtually constant velocity. Such phenotype was clearly inconsistent with classical equilibrium models and seemed indicative of a slow transition to the product-inhibited state. Mechanistically, this transition would comprise (fast) product binding and (slow) reorganisation of the two GGDEF domains to acquire the inactive product cross-linked configuration (Fig. 8).

The progress curves were fitted with the kinetic model shown in Fig. 9d. Independent binding of two substrate molecules (S) to the dimeric enzyme (EE) was parametrised with an equilibrium constant $K_d$ (assuming fast substrate binding), whereas the transition between active and inactive states was modelled

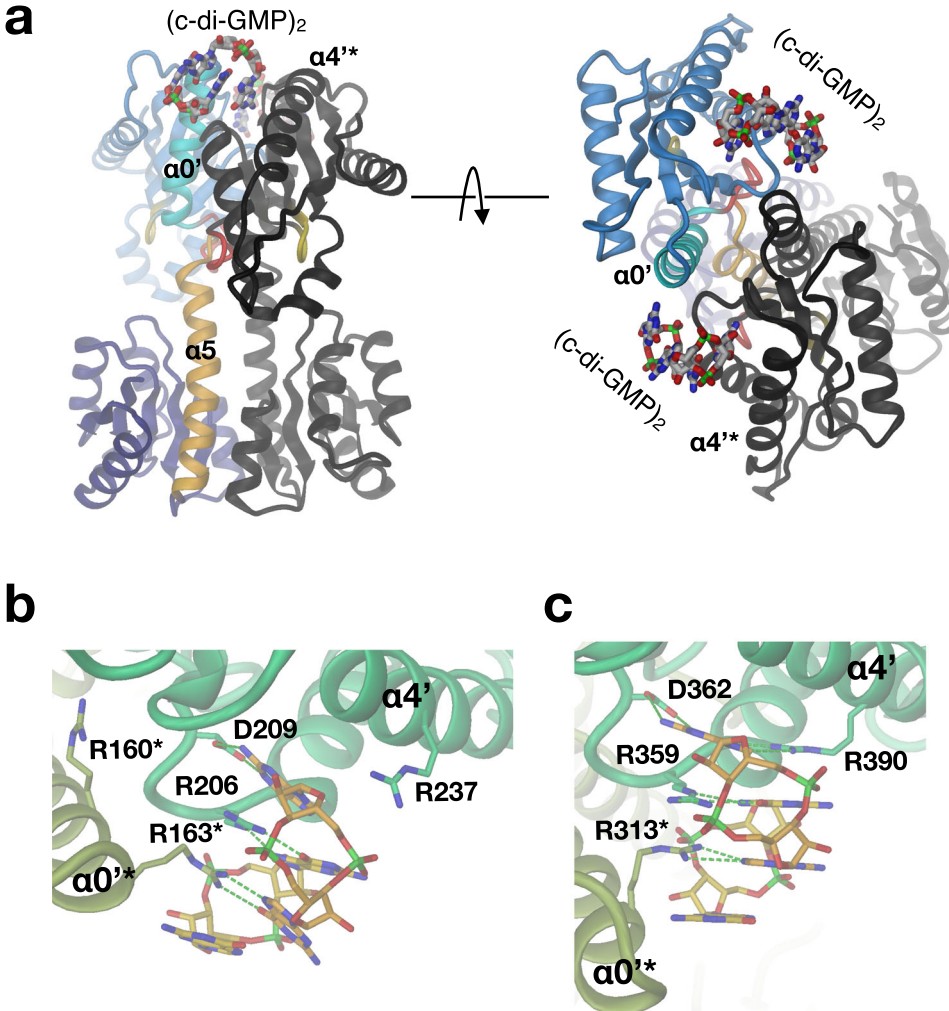

**Fig. 8 Dimeric c-di-GMP cross-links the GGDEF domain of the dimer. a** Side and top views of the DgcR/c-di-GMP complex (DgcR_inh). Representation as in Fig. 1b. **b**, **c** Detailed comparison of the c-di-GMP binding mode in DgcR (**b**) and PleD (2V0N) (**c**). Protomers are distinguished by colour hue. H-bonds are shown as green broken lines.

kinetically with an effective second-order rate constant $k_{on}$ (dependent on product and enzyme concentration) and a first-order rate constant $k_{off}$ with the inhibitory constant given by $K_i = k_{off}/k_{on}$. Note that for simplicity the model considers only one product binding site on the dimeric enzyme, while there are actually four (two c-di-GMP dimers). This simplification will affect the nominal value of $K_i$. Full kinetic modelling without this simplification and with explicit modelling of the conformational enzyme transition has been postponed to a follow-up study.

The kinetic model fits the biphasic curve of DgcR* very well (Fig. 9a, b) yielding the parameters given in Supplementary Table 2. The $k_{cat}$ of 0.33 s$^{-1}$ together with the slow kinetics of the active to inactive transition ($k_{off}$ about 10$^{-3}$ s$^{-1}$) explains the large build-up of product in the initial phase, which is followed by very low residual activity of the (equilibrated) sample due to the low $K_i$ of about 150 nM

To validate the involvement of the RxxD motif in feed-back product inhibition as suggested by the crystal structure (Fig. 8) and shown for many other DGCs, but also to scrutinise the kinetic model, the AxxA variant (DgcR') was analysed. The activated variant (DgcR'*) is highly active (Fig. 9b, c) and the progress curve can be modelled with the same $k_{cat}$, but a drastically (almost 50-fold) increased $K_i$ as compared to wild-type (Supplementary Table 2). Thus, as intended, the mutations do not

affect the catalytic efficiency of the activated enzyme, but render the enzyme largely insensitive to feed-back inhibition. Note that the mutations did not completely abolish inhibition, which may be explained by the remaining residues of the primary and secondary I-site (Fig. 8) still enabling (weak) product binding. The native variant showed a significantly lower activity than the native wild-type (Fig. 9b, c), but this difference in basal activity was not investigated further. Interestingly, for both wild-type and variant enzyme, the activated state was found to be more susceptible to product inhibition than the native states (Supplementary Table 2). A difference in the $K_i$ values is not surprising per se considering that formation of the back side cross-linked dimer should be influenced by the geometry of the dimeric Rec stem that is different in the two states.

Summarising, activated DgcR shows a pronounced initial burst of activity before entering the product-inhibited state with a rather slow kinetics probably reflecting domain reorganisation. The kinetic model (Fig. 9d) proved to reproduce all measured progress curves and the parameters (Supplementary Table 2) reflect the impact of activation and I$_P$-site mutation.

**Rec–GGDEF linker sequence profiles are consistent with register shift mechanism.** DgcR has been selected as a prototypic

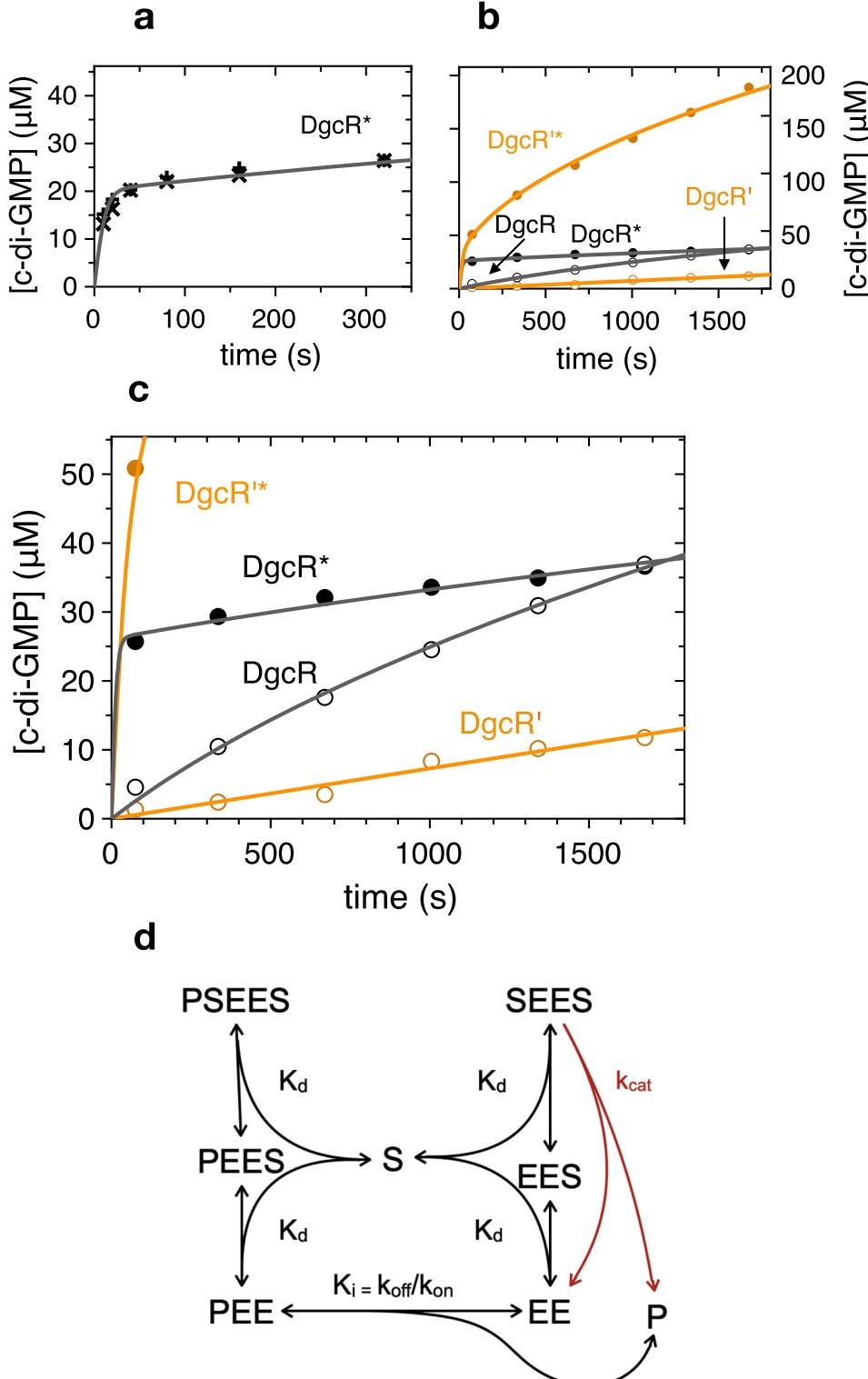

**Fig. 9 Enzyme kinetics of DgcR. a–c** Enzymatic progress curves of wild-type DgcR and inhibition relieved mutant DgcR' in the native and in the activated (indicated by asterisk) state. Experiments were performed with 5 µM enzyme and 500 µM GTP substrate concentrations. Symbols denote experimental values, continuous lines represent fit of the kinetic model shown in panel **d** to the data with parameters listed in Supplementary Table 2. **a** Progress curve of c-di-GMP production catalysed by DgcR* as measured by conventional IEC. **b** Progress curves as measured by oIEC catalysed with the indicated DgcR variants/states. **c** Zoom- in of **b**. **d** Kinetic model of diguanylate cyclase activity controlled by non-competitive product binding. Substrate (S) binding to the dimeric enzyme (EE) is modelled with the equilibrium dissociation constant $K_d$ and assumed to be unaffected by the presence of S in the second binding site or of product (P) in the allosteric site. Product binding is modelled kinetically with rate constants $k_{on}$ and $k_{off}$. Note that the model considers simply one instead of four product binding sites on the enzyme. Only the Michaelis–Menten complex with two bound substrate molecules and no bound product (SEES) is competent to catalyse the S + S → P condensation reaction (with turn-over number $k_{cat}$). Source data are provided as a Source Data file.

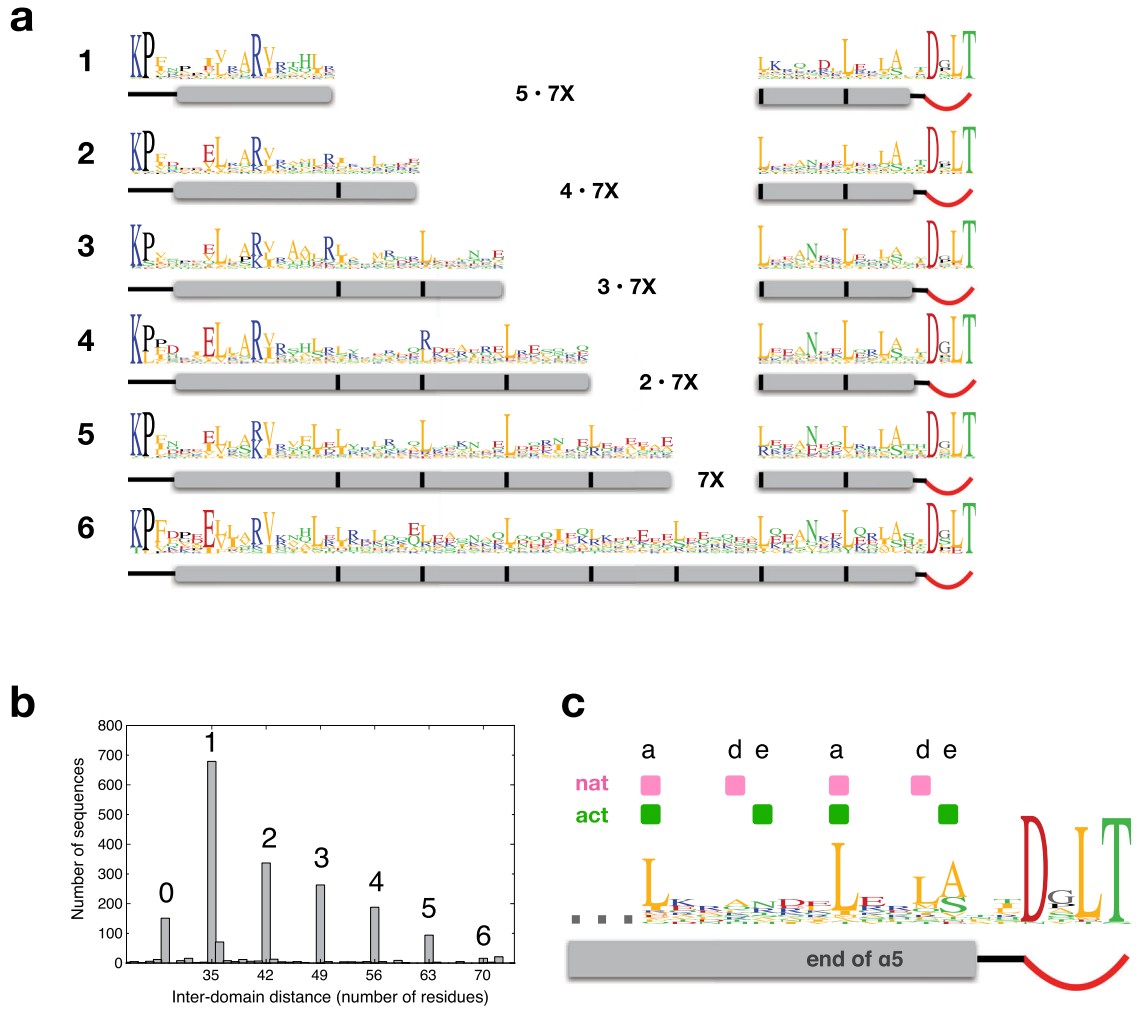

**Fig. 10 The linker helices of Rec-GGDEF proteins show heptad repeat patterns and discretized lengths. a** Sequence logos of inter-domain linkers grouped according to length. Logos 1 to 6 correspond to peaks 1 to 6 in the histogram of panel b. DgcR belongs to group 1. The grey rectangles symbolise the predicted α5 helices, black bars indicate recurring hydrophobic positions spaced with a distance of 7. Data were compiled from 1991 Rec-GGDEF sequences, see Methods. **b** Histogram of inter-domain distances as measured from Rec KP motif to GGDEF DxLT motif. **c** Overall logo of C-terminal part of all sequences shown in panel **a**. Positions engaged in parallel coiled-coil interactions in DgcR′ and DgcR′*, are indicated in pink and green, respectively (see also Fig. 5b).

Rec-GGDEF enzyme of relatively small size (298 residues), but bioinformatic analysis showed that the linker length can vary considerably in this class of DGCs. This was surprising considering that the linker has a defined structure and seems crucial for signal transduction. However, the linker length histogram (Fig. 10a) shows that the lengths are not distributed uniformly, but exhibit discrete values separated by multiples of 7 (groups 1 to 6, with DgcR and WspR belonging to groups 1 and 4, respectively). Thus, members of the groups would merely differ in the number of double helical turns when forming parallel coiled-coils. Indeed, the individual sequence logos can easily be aligned (Fig. 10b) to reveal the striking repeat of leucines in every 7th position (heptad position *a*). Most interestingly, the last (and to a lesser degree the last but one) heptad repeat at the C-terminal end (Fig. 10c) shows a conserved *axxdexx* pattern as in DgcR (Fig. 5). Thereby, the *a* positions are mainly occupied by Leu, while the *d* and *e* positions show more variation, but with most residue types (e.g., Ala, Ser, Asn) known to allow coiled-coil interaction. Probably, the variability in the conditional *d* and *e* positions reflects the requirement of weak interactions to allow conformational switching. Thus, a common binary register shift mechanism seems likely for members of all the groups. Group 0

(Fig. 10a, b) does not obey the linker length rule. Since it also has an (S/N)PLT instead of a DxLT motif, it probably has a different linkage and, therefore, activation mechanism.

A similar pattern of discretized coiled-coil lengths has been reported for PAS-GGDEF and LOV–GGDEF proteins[4,30], which makes it tempting to speculate that input to effector signal transduction might work similarly as in Rec-GGDEF enzymes. However, further investigations into their sequence profiles are needed to see whether they also exhibit ambiguous *axxdexx* heptad repeats.

**Conclusion**

The presence of coiled-coil linkers between N-terminal regulatory and catalytic GGDEF domains in many diguanylate cyclases has been described and their role in signal transduction discussed[4,9,13,30]. Changes in the crossing angle or the azimuthal orientation of the helices upon activation were anticipated, but a repacking of the interface was not discussed, which was then seen first in the comparison of inactive and a constitutively activated variant of light-regulated PadC[24]. The now presented detailed structural analysis of DgcR in its native and pseudo-phosphorylated form allowed a comprehensive dissection of the

activation process for a full-length, wild-type Rec-GGDEF enzyme (Fig. 7). Tertiary and quaternary changes in the Rec input domains lead to a register shift in the coiled-coil linker repositioning the inter-domain hinge and, thus, the propensity of the GGDEF domains to attain the catalytically competent dimer constellation.

A register shift in the coiled-coil linker may be operational also for other enzymes with predicted coiled-coil linkers, e.g., DGCs with N-terminal GAF domains or trans-membrane helices. LOV sensor domains that carry a flavin-nucleotide chromophore and have been studied very well as part of HKs[4,31] are different in that the coiled-coil forming Jα helix is not part of the core fold, but rather an extension of the C-terminal Iβ-strand that projects outward in the same direction. It has been shown that, upon light activation, the two Iβ-Jα junctions of the dimer increase their distance considerably[32] probably causing a change in the crossing angle and/or the super-twist of the Jα coiled-coil in the full-length protein to control activity as discussed in the recent review by Möglich[31]. Most likely, GAF domain proteins control GGDEF activity in a similar way, due to the structural similarity with LOV, including the predicted C-terminal coiled-coil[30]. HAMP domains have been shown to operate as rotary switches[33]. How such a change will affect the geometry of the C-terminal coiled-coil in respective DGCs has not been studied, but it will surely affect the relative disposition of the hinges that lead to the catalytic domains and, thus, activity.

Apparently, the coiled-coil linker is a versatile and effective means of transmitting a signal between domains without requiring direct interactions between them, which, obviously, is of paramount advantage for their modular combination in evolution. The same principle seems to apply also for HKs, many of them both are controlled by the same kind of input domains as DGCs and exhibit a coiled-coil preceding the DHp α1 bundle[34,35]. Signalling, however, seems to proceed via helix rotation[36,37] or depend on non-canonical coiled-coil geometry[38] and does not invoke lateral helix translation as found here for DgcR. Bioinformatic analyses[4,39] may now be extended to test for the occurrence of "slippery" heptad repeats in coiled-coil proteins in general to reveal proteins potentially signalling via coiled-coil register shifts.

## Methods

**Protein expression and purification**. *E. coli* BL21 (DE3) cells transformed with pET-28a vector containing DgcR full-length construct purchased from Genscript Inc. were incubated at 37 °C with agitation until they reached the optical density of 0.8–0.9. Expression was then induced by the addition of IPTG (Isopropyl β-D-1-thiogalactopyranoside) at a final concentration of 400 μM for 4 h at 30 °C. The cells were harvested after centrifugation and resuspended in a buffer composed by 20 mM Tris pH 8.0, 500 mM NaCl, 5 mM MgCl₂, 5 mM 2-Mercaptoethanol and protease inhibitor (Roche). The lysis proceeded by 3 passages in a French press cell at a pressure of 1500 psi. After a centrifugation at 30,000 x *g* for 50 min, the soluble fraction was loaded onto a His Trap HP 5 mL column (GE Healthcare) in 20 mM Tris pH 8.0, 500 mM NaCl, 5 mM MgCl₂, 5 mM 2-Mercaptoethanol and 20 imidazole. DgcR was eluted using imidazole gradient of 20 mM to 500 mM in 15 column volumes. The fractions containing DgcR were further purified by size exclusion chromatography using a Superdex 200 26/600 column (GE Healthcare) in 20 mM Tris pH 8.0, 20 mM NaCl, 5 mM MgCl₂ and 1 mM DTT. As comparison, protein was also purified using S75 16/60 and S200 16/60 (GE Healthcare) and showed an improved purification profile when using S 75 16/60 (Supplementary Fig. 3). Protein was quantified using a NanoDrop 2000 spectrophotometer (Thermo Fisher Scientific). The same expression and purification procedure was applied for wild type and I-site variant (R206A/D209A; DgcR'). DgcR' was produced by site-directed mutagenesis with the primers listed in Supplementary Table 3.

**BeF₃⁻ modification of DgcR**. In order to produce BeF₃⁻-modified DgcR, ~300 μM of DgcR in 20 mM Tris pH 8.0, 20 mM NaCl and 5 mM MgCl₂ were incubated with a mixture of NaF at 10 mM and BeCl₂ at 1 mM, final concentration. After gentle mixing to achieve a homogeneous solution, the sample was left at room temperature for at least 15 min. DgcR BeF₃⁻ mix was then centrifuged at 4 °C at 18,000 x *g* to remove a light precipitation formed during the process. Protein concentration was measured after the activation process and was found virtually unaltered.

**SEC-MALS analysis**. Light-scattering intensity and protein concentration were measured at elution from the column using an in-line multi-angle light-scattering and differential refractive index detectors (Wyatt Heleos 8+ and Optilab rEX). These data were used to calculate molar mass for proteins by standard methods in Astra 6 (Wyatt). Corrections for band-broadening, inter-detector delays and light-scattering detector normalisation were performed using a sample of bovine serum albumin in the experimental buffer, according to the manufacturer's protocol. Samples were loaded (100 μL) at concentrations ranging from 0.5 to 10 mg/mL in presence of various ligands at a constant flow of 0.5 mL/min in 20 mM Tris pH 8.0, 500 mM NaCl, 5 mM MgCl₂, 1 mM DTT or same buffer but with 20 mM NaCl.

**Crystallisation**. Crystallisation attempts were performed using vapour diffusion method prepared in 3-drop MRC plates by Gryphon robot (Art Robbins Instruments) with DgcR (wild-type or I-site variant AxxA) at a concentration of 10 mg/mL (280 μM) in 20 mM Tris pH 8.0, 20 mM NaCl, 5 mM MgCl₂ and 1 mM DTT at 18 °C. For DgcR' crystallisation, 3'dGTP was added at a final concentration of 2 mM. After 3 days, crystals could be observed in 0.2 M Magnesium sulphate, 20% PEG 3350 from condition C8 of PEG/Ion HT crystallisation kit (Hampton Research). DgcR'* was crystallised by the same protocol, but with BeF₃⁻ treatment prior to the crystallisation set-up. After 7 days crystals were observed in a condition composed by 0.3 M Magnesium chloride hexahydrate, 0.3 M calcium chloride dehydrate, 1.0 M imidazole, MES monohydrate (acid), pH 6.5, EDO_P8K, 40% v/v ethylene glycol, 20% w/v PEG 8000 present in condition A2 from Morpheus I crystallisation kit (Molecular Dimensions). Crystallisation of DgcR in the inhibited conformation (DgcR_inh) was achieved by the presence of 2.0 mM c-di-GTP. Crystals appeared after 5 days in 0.2 M potassium thiocyanate, 0.1 M Tris pH 7.5, 25% PEG 2000 MME, condition optimised from H11 of Index HT crystallisation kit (Hampton Research). Crystals were frozen in liquid nitrogen and stored in a transport Dewar prior to data collection.

**Crystal data collection and structure determination**. Data was collected at the Swiss Light Source (SLS), Villigen, Switzerland at 100 K with DA + data acquisition software and was processed using XDS (DgcR'), iMosflm (DgcR'* data and DgcR_inh) and CCP4i suite[40–42]. DgcR' structure was solved by molecular replacement using homologous structures generated from the Auto-Rickshaw pipeline web server[43]. Subsequently, the DgcR'* and DgcR_inh structures were solved by molecular replacement using the Rec and GGDEF domains of DgcR' separately using Phaser[44]. The model was built using COOT and refinement was carried using Refmac5 using local NCS restraints[45,46]. Structure figures were prepared using Dino (http://dino3d.org). Morphing was calculated using UCSF Chimera[47].

**Enzymatic analysis**. DgcR wild type and DgcR_AxxA (DgcR') activity assays were performed at 5 μM in the presence of 500 μM of GTP in a reaction buffer composed of 100 mM Tris pH 8.0, 100 mM NaCl and 5 mM MgCl₂. The reaction was started by substrate and product progress curves were acquired by automatic chromatographic method, named online ion-exchange chromatography (oIEC) (Agustoni, manuscript in preparation), in which aliquots (68 μL) are automatically withdrawn from the large reaction vessel (650 μL) and loaded into a Resource Q column (GE Healthcare) without the need for prior quenching of the reaction. This was followed by ammonium-sulphate (0 to 1 M, 20 mM tris, pH 8.0) gradient elution of the bound substances (enzyme, substrate, product). Peak areas corresponding to the c-di-GMP product were integrated and converted to concentrations using a scale factor obtained from calibration. To check for intermediate formation, 10 mM HCl was used as elution buffer with an NaCl gradient 0 – 400 mM. Data was plotted and fitted using proFit (QuantumSoft). To calculate theoretical progress curves, the partial differential equations corresponding to the kinetic scheme in Fig. 9b were set-up in ProFit and solved by numerical integration. Global fitting of this function using the Levenberg algorithm implemented in ProFit to the measured time courses of product and substrate concentration yielded the parameters listed in Supplementary Table 2.

**Bioinformatic analysis**. Rec and GGDEF domain HMM profiles were taken from Pfam[48] and used as input to an hmmsearch on the HMMER web server against the reference proteome database rp55 (*E*-values 0.01; hit 0.03)[49]. 8016 sequences were found and filtered by size (<360 residues) to exclude Rec-GGDEF sequences with additional domains. This procedure reduced the data size to 1991 sequences. A redundancy filter (<80% pairwise identity) finally reduced the number of sequences to 1408. Global alignment was performed using Muscle[50]. From this alignment, the linker sequences (as defined ranging from the KP-motif in the Rec β5-α5 loop to the DxLT motif at the beginning of the GGDEF domain) were extracted and clustered according to length by a custom-made Python script. For the major clusters, corresponding logos were generated using Geneious Prime 2020.1.2 (www.geneious.com) and manually aligned to account for the distinct linker lengths.

**Reporting summary**. Further information on research design is available in the Nature Research Reporting Summary linked to this article.

## Data availability

Coordinates and structure factors of DgcR', DgcR'*, and DgcR_inh have been deposited in the Protein Data Bank (https://www.rcsb.org) under the, respectively, accession codes 6ZXB, 6ZXC and 6ZXM. Pfam database was accessed at https://pfam.xfam.org. Source data are provided with this paper.

## Code availability

The function used for fitting kinetic data was added to https://github.com/teixeirard/ncpi_2S.git.

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

## Acknowledgements

We thank the beamline staff at the Swiss Light Source in Villigen and the Biophysics facility at the Biozentrum Basel for expert biophysical support. We thank T. Sharpe,

E. Agustoni, U. Jenal, and T. Maier for critical reading of the manuscript. This work was supported by Grant 31003A-166652 of the Swiss National Science Foundation.

## Author contributions

R.D.T., F.H. and T.S. designed the experiments; R.D.T., F.H. performed the experiments; and R.D.T. and T.S. interpreted the results and wrote the manuscript.

## Competing interests

The authors declare no competing interests.
