## [Peer Review File · Nature Communications]

REVIEWER COMMENTS

Reviewer #1 (Remarks to the Author):

The manuscript by Teixeira et al. describes an interesting multidomain system that integrates a phosphorylation event at the receiver domain to activate a covalently linked diguanylate cyclase. The authors report crystal structures in a non-activated as well as in an activated receiver state and observe characteristic structural rearrangements at the coiled-coil domain linker. These structural changes are apparently required to enhance conformational sampling of the diguanylate cyclase domains to facilitate productive encounter of two GTP molecules.

Based on their results the authors have also performed a bioinformatic analysis of other receiver-diguanylate cyclase systems and observed an interesting conservation of linker lengths and compositions. With respect to the composition conservation, the observed coiled-coil register switch involved in enzyme activation appears to be operational in most of these systems, as well as in other sensory diguanylate cyclases (as described previously for some of these systems).

This is a very well performed study with interesting insights into the Rec-GGDEF system, but also with more general implications for GGDEF regulation. It extends previous descriptions of register switching in coiled-coil linkers by showing this conformational switch directly for the pseudoactivation of the sensor domain. Assuming that the applied substitutions of the inhibitory site do not drastically change the regulatory mechanism, this study also shows the coiled-coil switching in a near native environment.

All in all, this is a nice and thoroughly conducted characterization of an interesting sensor-effector system. To further improve the quality of the manuscript, I would recommend to put some more effort into the following points. While these are mostly minor issues, there are two points that in my opinion require more efforts:

- the kinetic characterization; GTP to c-di-GMP conversion involves a complex and not completely understood mechanism of substrate binding, sometimes conversion to a linear intermediate, product formation, product dissociation and potential rebinding and inhibition. The authors mention a new detection system for substrate and product, but do not refer to any intermediate production. Considering that this is not produced in DgcR, it would be worth mentioning this. It should also be stated in the methods that substrate, product and intermediate can at least be detected separately - if not this might have consequences for the interpretation of the kinetics and the discussed mechanism.

In fact, also the current discussion of why activated wild-type has a stronger product inhibition than non-activated wt could be extended. Similarly, the relatively strong impact of the inhibitory site substitutions on enzymatic activity are not discussed in the detail that one might expect since all the structures were obtained for the inhibitory site variant.

Also I would recommend to include some statistics for the kinetic measurements. Especially since the pseudo-activated wt and variant measurements show a strong burst phase, I would appreciate some indication of the reproducibility (including also blank reactions to confirm that this is no artefact of the BeF3- treatment).

Also with respect to the fitting to kinetic models, one might expect to see more data points in the burst phase that support the kinetic modelling as performed.

- the structure refinement description; looking at the validation reports for the deposited structures, there are unexpectedly many RSRZ outliers for some of the structures. Especially for chain D in one of the structures. Maybe the readers could benefit from some remarks as to potential issues during the refinement process. Another issue might be the GH3 ligand; is the restraint file used by the authors

incorrect or the deposited geometries for this entry? With even some chirality outliers and very strong clashes in certain sidechains, I would recommend to double check the final deposited structures.

For the structure with 6 molecules in the asymmetric unit the number of unique reflections to parameters ratio approaches a region where probably many restraints had to be imposed during the refinement. These should be mentioned in the methods section.

minor observations:

line 24 - contain

line 77 - in general I find DgcR_nat misleading for the variant; especially since this is also not consistently used later when comparing to the wild-type. But this is a personal opinion ...

line 101 - the end? more the beginning or the N-terminal part of alpha0

line 171 - bracket missing

line 444 - Refmac or PHENIX (as in the validation reports)

Fig. 3 - Tln DgcR_act ?

Supp Fig. 1 - I am pretty confident that the authors have looked at this, but in the superimposition it looks as if aligning the respective other protomers could allow a better overall fit!?

Reviewer #2 (Remarks to the Author):

Teixeira & coll. disclose three crystal structures of the diguanylate cyclase (DGC) DgcR from *Leptospira biflexa*.

This is a fine piece of work, technically sound and well written.

There are a few typos here and there that probably deserve a careful double check. Background literature is well referenced and up to date.

Very few 3D structures of full-length, multi-modular DGCs, have been reported to date, increasing the importance of studies such as this one. The Schirmer lab has a long-standing interest in DGCs, having pioneered the field with work on full-length PleD (comprising a REC-REC*-GGDEF domain architecture) from *Caulobacter crescentus*, and later the *E. coli* Zn-binding DgcZ (CZB-GGDEF).

Very few additional structures have been later determined from other labs, notably including WsprR from *P. aeruginosa* and PadC from *Idiomarina* sp.

The precise molecular mechanism(s) by which regulatory domains modulate GGDEF-mediated DGC activity is however still a matter of discussion, and extremely relevant due to the impact of c-di-GMP homeostasis in Biology.

In this ms Teixeira et al. set out to respond whether their previously proposed 'chopstick' model (Schirmer *J Mol Biol* 2016) is universally valid in sensor-regulated DGCs.

The choosing of a short full-length DGC, comprising a single regulatory domain (in this case a phosphorylatable REC) and a catalytically competent GGDEF, both connected through a coiled-coil (CC) segment, facilitated the structural and functional characterization.

All in all, I believe this work will be of great interest for a wide audience within the communities of structural biologists, c-di-GMP biologists and microbiologists. A novel way of thinking about coiled-coil reorganization in signalling proteins, namely by discrete switching between two alternative heptad patterns, is being disclosed for the first time.

Major concerns

1- The authors conclude that DgcR is a constitutive dimer. This is important, because REC

phosphorylation could in principle favor dimerization and thus raise the effective concentration of the two GGDEFs, ultimately increasing DGC activity. If so, this would correspond to the known 'PleD-kind' of mechanism.

MW estimations to characterize DgcR's quaternary structure are thus critically important, scrutinizing whether there are monomer-dimer or dimer-tetramer equilibria. Key parameters to do so are not clear enough to me, so I do have a few questions:

- what are the results in SEC-MALS when using 20mM NaCl? (20mM NaCl seems to work OK in your purification protocol). The reported 500 mM NaCl for SEC-MALS (Mat & Meth refers to 500 mL, but I assume this is a typo), is a quite high salt concentration. High ionic force will reinforce hydrophobic-mediated interactions, and potentially hold monomers together in a more stable dimeric form, even at low protein concentrations.
- please share the chromatograms corresponding to the SEC purification step (an additional Supplementary figure would be helpful)
- why was a Superdex 200 column chosen, instead of an S75? I suggest repeating the separation with the latter one (at difference with S200, in an S75 the monomeric and dimeric species are expected to be included and separated with greater resolution). A potential tetramer would elute within the exclusion volume, neatly distinct from dimers and monomers.
- how does the AxxA mutant behave in SEC-MALS? (I understand from Mat & Methods that only the wild-type protein was used) Can different quaternary structures for wild-type vs AxxA be ruled out based on direct evidence?

2- Precisely how are crystal packing contacts fixing in place the GGDEF domains in the native and activated AxxA structures? The authors do say that the GGDEF domains are being held in position by crystal contacts (in both structures), I believe this deserves a more elaborate description (including with an illustrative supplementary fig). This comment leads to two connected issues:

* more evidence appears to be needed to substantiate the claim that the AxxA native structure represents a catalytically non-productive configuration, whereas the AxxA activated one is poised for catalysis (as an immediately previous state to the catalytically competent one; illustrated in Fig 7 in the transition from panels e to f).

If the GGDEF domains are indeed extremely flexible relative to the rest of the molecule, how reliable is the GGDEF orientation information, comparing the native and activated structures, to interpret steps along the activation process (fig 7)?

* could the role of REC phosphorylation in modulating DGC activity be further elaborated? This issue interrogates the section on "Structural coupling of Rec modification with competent dimer formation" (page 11-12).

An apparently subtle difference seems to me a very important one, comparing figures 1b vs 7a. The unphosphorylated, native form of DgcR (Fig 1b and 2b,c) is actually proving that the GGDEFs can adopt a configuration poised to adopting a competent constellation, namely the one adopted by chain B (not included in Fig 7a).

The authors emphasize that there are almost no direct constraints between REC and GGDEF domains, the latter enabled to move quite freely. So, at the end of the day the question seems to remain unanswered. Please clarify.

3- How come the c-di-GMP Ki figures are different for native vs active species? This appears to indicate that REC phosphorylation is engaged in modulating REC-GGDEF interaction. The authors briefly refer to suboptimal product-mediated cross-linking when the protein is not phosphorylated, but in any case, this suggests that GGDEF domains are not really independent (which contradicts the authors' statement). Please correct and/or clarify in the main text.

Minor comments

- The Abstract would improve by explicitly stating the knowledge gap that the authors set out to address. A phrase in this sense is expected separating the initial background sentences, and the actual results that are being reported now.

- Substitute reference #15 by the original source of morbidity/mortality burden : Costa et al. Global morbidity and mortality of leptospirosis: a systematic review. PLoS Negl Trop Dis 2015;9:e0003898.

- Following up on the previous point, ref #16 might also be substituted by a more general review wherein using *L. biflexa* as a model is also mentioned; for example, Picardeau Virulence of the zoonotic agent of leptospirosis: still terra incognita? Nat Rev Microbiol 2017 15:297.

- Previous work on histidine kinases is referred to by the authors as an inspiring source of information concerning mechanisms of signal transmission via coiled-coils. I candidly believe that at least two key reports from our group deserve fair credit, because they are pertinent in the context of the authors' discussion and would likely help to enrich its views and better put the work in context.

*Our review (Buschiazzo & Trajtenberg Annu Rev Microbiol 2019) should probably be cited when you invoke HK phosphorylation of REC domains in the Introduction, as one of the most updated reviews in the field. Apart from alternative heptad registers, on which the authors pertinently focus on, this review will likely help in enriching the Discussion, bringing up the universal feature of non-ideal heptads (i.e. including polar residues at a/d positions, and heptad phase adjustments such as skip positions, stutters and stammers). Such departure from ideal hydrophobic packing enables iso-energetic conformational rearrangements: indeed, Fig 10a in this ms readily shows the presence of polar residues like Thr and Arg at some 'a' positions, and even more diverse variations at 'd' positions. * The authors here refer to examples of coiled-coil signal-transmission in HKs, citing Diensthuber et al Structure 2013 and Wang et al PLoS Biol 2013 (line 388). However, our work on the HK DesK (Albanesi et al PNAS 2009) seems also particularly relevant, as it was the first experimental report showing the workings of signal-transmission via coiled-coil signaling helices in HKs, which had previously been predicted (Anantharaman et al Biol Direct 2006) Our work came out months after the Moffat lab presented convergent results, albeit working with chimeric LOV-HK constructs (Möglich et al 2009 J Mol Biol). Albanesi et al 2009 was followed by Saita et al Mol Microbiol 2015 and Trajtenberg et al eLife 2016, wrapping up one of the most complete examples of coiled-coil mechanistic studies in HK signaling.

- The Methods section says that the minimum concentration of protein analyzed by SEC-MALS was 0.4 mg/mL, but figure S2 shows it was 0.5 mg/mL. Which one is correct? Moreover, Fig S2's legend refers to 0.014 mM as the lowest protein concentration, from what I deduce the authors consider the species to be a monomer (corresponding to 0.007 mM of dimer), yet showing it behaves as a dimer in SEC under these conditions.

- Fig S2 caption does not describe how much BeF3- was used.

- I can only guess that crystallization of wild-type DgcR didn't work, why? considering that the AxxA variant is used as a good model of the wt, substantial structural variations between them are not expected, right?

Why was 3'dGTP added? What if it is not added?

- The phosphorylation-induced reorganization in the REC domains, shifting the interaction between a4-b5-a5 faces, seems to be very similar to the one that was observed between the REC and pseudoREC domains in PleD -in which case this favored subsequent dimerization. Is this correct?

I wonder whether without the CC, DgcR's REC domains alone would otherwise undergo a P-triggered monomer-to-dimer equilibrium shift. Have you envisaged a REC-only construct of DgcR, to evaluate this? If a REC-only monomer to dimer shift were confirmed due to phosphorylation, this could be important in modifying the dynamics even if the full-length dimer is indeed constitutive (HDX-MS -

such as done with IsPadC- and/or SAXS would also be confirmatory of such scenario)

- Fig 2: the plot in panel (a) is not that clear: it would be helpful to show nat_A (to readily compare to chain B); the boxes do not correspond straightforwardly to residues on the x-axis

- Fig 3: the caption indicates Thr86, but in the text this Thr is first numerated as 85 (page 5 line 119) I guess the latter is a typo, afterwards in the text (line 137) Thr86 is mentioned. A closeup into the phosphorylation site would be very useful to see the shifts (e.g. adding a panel c).

- Fig 6 : I can't fully understand what you are plotting in panel (a): what is exactly represented with the last two boxes? Is it torsion angles for residue 137? Or yet the manually produced shifts on residue 136 phi/psi angles? The x-axis is confusing

Alejandro Buschiazco, PhD

Reviewer #3 (Remarks to the Author):

Diguanylate cyclases (DGC) catalyze the formation of the widespread bacterial second messenger c-di-GMP. DGCs, also referred to as GGDEF, act as obligate dimers with the active site formed across the dimer interface. The dimeric DGCs are found in conjunction with a palette of different N-terminal sensor/input domains to which they are connected by dimeric, parallel α -helical coiled coils. The architecture REC-GGDEF studied presently by Schirmer and colleagues comprises an N-terminal receiver domain and is particularly common in nature. REC domains form part of two-component systems and are phosphorylated by histidine kinases to regulate the activity of associated output modules. In the present contribution, the authors report crystal structures of DgcR from *Leptospira biflexa* in its native form, a pseudo-activated form (in the presence of BeF_3^-), and a product-inhibited form. The detailed comparison of the structures allow the authors to infer a likely molecular trajectory via which DgcR is activated upon phosphorylation. A rearrangement of the REC dimer interface is relayed to and through the coiled coil as a change in coiled-coil register. This structural transition arrives at the GGDEF dimer and enables the adoption of the catalytically competent state. The product inhibition is apparently based on c-di-GMP-mediated inter-protomer contacts that capture the GGDEF dimer in an unproductive arrangement.

Taken together, this is an equally well conceived and executed manuscript that reports an abundance of relevant and captivating data; easily the best manuscript I have read in many weeks. The experiments have been executed to the highest standard, and likewise the manuscript is written clearly and carefully. All major conclusions are fully supported by the data. I strongly expect the work to be of profound interest to a broad readership, and I hence recommend publication in *Nat Commun* after suitable revision. Most comments I have are relatively minor and are listed below, except for two aspects: first, the native structure of DgcR shows asymmetry of the GGDEF domains but this observation is almost brushed aside. In fact, in the mechanistic model in Fig. 7 the dimeric molecule has been symmetrized. But could it be that the observed dimer structure is not just a product of crystal packing but has biological significance? As the authors will know, asymmetry abounds among the structures of sensor histidine kinases and has been ascribed functional relevance. I encourage the authors to entertain this idea for DGCs and remark on it in the Conclusion, even should this amount to speculation. Second, the kinetic modeling shown in Fig. 9 should be revised: in my understanding, the arrow labeled ' k_{cat} ' should originate at SEES but then point towards P (as it currently does) and EE (rather than EES). If this assessment is correct, then the modeling should be updated accordingly.

Minor comments:

- 34: insert 'molecules' after 'GTP'

- 45: insert 'PAS-GAF-' before 'PHY'

- 70: '... dimer that is activated ...'

- 94: it would be informative to report/show the orientation of the screw axis around which the GGDEF protomers are related by 90° . Is this axis coincident with the C2 axis for the REC and coiled-coil

portions?

- 114: abbreviation MALS should be spelled out
- 216-231: some more information on how the catalytically competent conformation was generated is warranted. Presumably this state was modelled based on a previously determined DGC structure which should be mentioned here.
- 258: see above general comment, how/why was the structure symmetrized? Is there any indication that asymmetry may play a role?
- 265-270: why do the GGDEF domains not adopt the catalytically competent arrangement in the native state? Would a steric clash result in this arrangement?
- 308: can a timescale be provided for the statement 'slow transition'?
- 312-324: in the scheme in Fig. 9, it would help if some of the arrows were fused in the manner done for the arrow denoted 'kcat'. At first glance, I found the scheme quite confusing but suspect that it may be clearer if this change was accommodated. And, as noted above, it appears that the arrow 'kcat' should point to EE as opposed to EES, no?
- 321-324: what is the confidence on the parameter values?
- 328: rephrase 'inhibition relieved'
- 332: insert 'basal' before 'activity'
- 352: elaborate on the conserved axxdexx pattern. What is the consensus sequence for this pattern, and how does it support the alternate coiled-coil registers?
- fig. 9: it is rather surprising that at long times the non-activated wild-type DGC exhibits higher turnover than the activated one. What is the (suspected) origin of this observation?

Reviewer #1 (Remarks to the Author):

The manuscript by Teixeira et al. describes an interesting multidomain system that integrates a phosphorylation event at the receiver domain to activate a covalently linked diguanylate cyclase. The authors report crystal structures in a non-activated as well as in an activated receiver state and observe characteristic structural rearrangements at the coiled-coil domain linker. These structural changes are apparently required to enhance conformational sampling of the diguanylate cyclase domains to facilitate productive encounter of two GTP molecules.

Based on their results the authors have also performed a bioinformatic analysis of other receiver-diguanylate cyclase systems and observed an interesting conservation of linker lengths and compositions. With respect to the composition conservation, the observed coiled-coil register switch involved in enzyme activation appears to be operational in most of these systems, as well as in other sensory diguanylate cyclases (as described previously for some of these systems).

This is a very well performed study with interesting insights into the Rec-GGDEF system, but also with more general implications for GGDEF regulation. It extends previous descriptions of register switching in coiled-coil linkers by showing this conformational switch directly for the pseudoactivation of the sensor domain. Assuming that the applied substitutions of the inhibitory site do not drastically change the regulatory mechanism, this study also shows the coiled-coil switching in a near native environment.

We thank the reviewer for the very positive feed-back and the constructive comments, which refer mainly to the kinetic characterization and structure refinement. Our point-to-point response is given below (in blue).

All in all, this is a nice and thoroughly conducted characterization of an interesting sensor-effector system. To further improve the quality of the manuscript, I would recommend to put some more effort into the following points. While these are mostly minor issues, there are two points that in my opinion require more efforts:

the kinetic characterization; GTP to c-di-GMP conversion involves a complex and not completely understood mechanism of substrate binding, sometimes conversion to a linear intermediate, product formation, product dissociation and potential rebinding and inhibition. The authors mention a new detection system for substrate and product, but do not refer to any intermediate production. Considering that this is not produced in DgcR, it would be worth mentioning this. It should also be stated in the methods that substrate, product and intermediate can at least be detected separately - if not this might have consequences for the interpretation of the kinetics and the discussed mechanism.

The reviewer called attention to an important point, but there was no indication of another species in any of the chromatogram peaks. Nevertheless, we have run some more reactions with another elution solution (HCl) that we knew from other experiments gave better resolved peaks. All peaks that were obtained under the conditions that were used for Fig. 9 were clean. They can be seen in panels c - f of the new Suppl. Fig. 7.

The reaction intermediate indeed co-elutes with the product in our method that uses a tris-buffer for nucleotide separation. So, to address the question we re-ran the reactions and eluted the nucleotides with an acidic solution (10 mM HCl with 0 – 400 mM NaCl gradient) that we had observed before to separate c-di-GMP from the intermediate pppGp.

We do not observe intermediate formation under the reaction conditions used to determine kinetic parameters (Fig.9), i.e. at 5 μ M protein concentration. But interestingly we found formation of intermediate (eluting close to GTP) when using non-activated DgcR' (old name DgcR_nat, we have changed nomenclature, see further down) at high concentrations (30 μ M), see Suppl. Fig. 7g. The intermediate builds up only under conditions where the

enzyme is very ineffective, i.e. the second condensation would not follow before the intermediate leaves the enzyme. We have added these results to Supplementary Fig. 7 and also added information to Materials and Methods section.

Similarly, the relatively strong impact of the inhibitory site substitutions on enzymatic activity are not discussed in the detail that one might expect since all the structures were obtained for the inhibitory site variant.

We thank the referee for this valid comment that prompted us to improve the pertinent discussion, see lines 342 to 344. We now stress that the initial velocity (and thus the k_{cat}) of the activated mutant is comparable to that of the activated wild-type. Therefore, the structure of DcgR^{*} should be a very good model for DgcR^{*}. We see a difference in the k_{cat} values for the native proteins, but these k_{cat} values are indeed very low and represent a side-reactivity, which we did not investigate further. Furthermore, we added a comment about the difference in the K_i values for the two activated states (line 353, "A difference in the K_i values is not surprising per se").

Also I would recommend to include some statistics for the kinetic measurements. Especially since the pseudo-activated wt and variant measurements show a strong burst phase, I would appreciate some indication of the reproducibility (including also blank reactions to confirm that this is no artefact of the BeF₃⁻ treatment). Also with respect to the fitting to kinetic models, one might expect to see more data points in the burst phase that support the kinetic modelling as performed.

We agree that the burst phase is a very unexpected feature (that may be of relevance also for other DGCs). Therefore, we acquired data for the early time-points of the DgcR^{*} progress curve by conventional, i.e. manually EDTA stopped IEC (Fig. 9a). Since for this measurement we have used a new DgcR batch, we also had to remeasure the oIEC curves of DgcR and DgcR^{*} (Fig. 9b,c). Both oIEC progress curves agree well as indicated by the resulting parameters in Tab. S2, and, comforting, also the new oIEC data show the DgcR^{*} burst.

We did a control without enzyme and see that BeF₃⁻ does not react with GTP (no GTP → c-di-GMP conversion) (Supplementary Fig. S7a-b).

the structure refinement description; looking at the validation reports for the deposited structures, there are unexpectedly many RSRZ outliers for some of the structures. Especially for chain D in one of the structures. Maybe the readers could benefit from some remarks as to potential issues during the refinement process. Another issue might be the GH3 ligand; is the restraint file used by the authors incorrect or the deposited geometries for this entry? With even some chirality outliers and very strong clashes in certain sidechains, I would recommend to double the check the final deposited structures.

We have further refined and scrutinised the three structures, updated the structures on the PDB and the resulting validation reports are attached.

DgcR_{nat}: Though this is the structure with the highest resolution (2.2 Å), it shows a relatively large RSRZ of 8.0 %. This can be attributed to the large mobility of the Rec domains, in particular of the B chain (mean B-factor of Rec domains is 100 Å², cf. with overall B of 68.9 Å²). Correspondingly, although the main-chain is well defined, the electron density of many side-chains of the Rec domains is weak or absent. We added a note to the main-text (line 81).

DgcR_{act}: The RSRZ is 4.7 %, not much worse than expected for a 2.8 Å structure. But, indeed, chain D has a poor RSRZ with 11 %. Again, as above, this can be attributed to a large overall mobility resulting in weak or absent electron density for several side-chains. However, as expected chain D obeys NCS very well (with its domains superimposing with

less than 0.6 Å on those of the other chains), which gives, apart from the good density of the main-chain, confidence in the correctness also of this chain.

DgcR_inh: This is a medium resolution structure (3.3 Å). Although some quality indicators (RSRZ, side-chain rotamers) are poor, the domains nicely obey NCS (rmsd < 0.5 Å). Based on the R_{free} and the quality of the c-di-GMP densities, there is no doubt that the structure with all its 6 chains is correct.

We agree that the GH3 ligands show significant deviation from ideal geometry values as defined in the PDB validation report. We have inspected this problem by looking closer into the CCP4 library entries of GH3 and GTP. Indeed, the GH3 entry was not entirely consistent with the GTP entry for the guanine base, so we generated a new GH3.cif file based on GTP.cif, since the ideal bond distances and angles of latter file agree very well with a published high-resolution small molecule structure (PMID: 5537152). Since, due to the imposed restraints, our refined ligand structures obey these values closely, there is no problem with our refinement. Rather, the problem appears to be the library used by the PDB validation service (apparently derived from a MOGUL analysis). E.g. the C6 - N1 bond distance should be 1.40 Å (GH3.cif) and not 1.33 Å (as listed as ideal value in the validation report), similarly the C5-C6-N1 angle is 111.4 and not 123.4, and the C6-N1-C2 is 124.4 and not 115.9. We have informed PDB about this problem.

For the structure with 6 molecules in the asymmetric unit the number of unique reflections to parameters ratio approaches a region where probably many restraints had to be imposed during the refinement. These should be mentioned in the methods section.

All structures have been refined with local NCS restraints. A remark concerning the imposed NCS restraints has been added to the Methods section (line 470).

minor observations:

line 24 – contain

Corrected

line 77 - in general I find DgcR_nat misleading for the variant; especially since this is also not consistently used later when comparing to the wild-type. But this is a personal opinion ...

We thank the reviewer for this comment. Indeed, the nomenclature could be misleading and not the same names are used for the structures and the kinetic results. Thus, we have changed the nomenclature throughout to:

- DgcR_nat  DgcR'
- DgcR_act  DgcR'*
- wt  DgcR
- wt*  DgcR*
- mut_AxxA  DgcR'
- mut_AxxA*  DgcR'*

line 101 - the end? more the beginning or the N-terminal part of alpha0

This has been corrected

line 171 - bracket missing

This has been corrected

line 444 - Refmac or PHENIX (as in the validation reports)

Refmac was used, it has been corrected

Fig. 3 - Tln DgcR_act ?

This typo has been corrected

Supp Fig. 1 - I am pretty confident that the authors have looked at this, but in the superimposition it looks as if aligning the respective other protomers could allow a better overall fit!?

We have looked into this. The shown superimposition has an overall rmsd on C α of 8.9A while with the other promoters the rmsd is 10.5A.

Reviewer #2 (Remarks to the Author):

Teixeira & coll. disclose three crystal structures of the diguanylate cyclase (DGC) DgcR from *Leptospira biflexa*.

This is a fine piece of work, technically sound and well written.

There are a few typos here and there that probably deserve a careful double check. Background literature is well referenced and up to date.

Very few 3D structures of full-length, multi-modular DGCs, have been reported to date, increasing the importance of studies such as this one. The Schirmer lab has a long-standing interest in DGCs, having pioneered the field with work on full-length PleD (comprising a REC-REC*-GGDEF domain architecture) from *Caulobacter crescentus*, and later the *E. coli* Zn-binding DgcZ (CZB-GGDEF).

Very few additional structures have been later determined from other labs, notably including WspR from *P. aeruginosa* and PadC from *Idiomarina* sp.

The precise molecular mechanism(s) by which regulatory domains modulate GGDEF-mediated DGC activity is however still a matter of discussion, and extremely relevant due to the impact of c-di-GMP homeostasis in Biology.

In this ms Teixeira et al. set out to respond whether their previously proposed 'chopstick' model (Schirmer *J Mol Biol* 2016) is universally valid in sensor-regulated DGCs.

The choosing of a short full-length DGC, comprising a single regulatory domain (in this case a phosphorylatable REC) and a catalytically competent GGDEF, both connected through a coiled-coil (CC) segment, facilitated the structural and functional characterization.

All in all, I believe this work will be of great interest for a wide audience within the communities of structural biologists, c-di-GMP biologists and microbiologists. A novel way of thinking about coiled-coil reorganization in signalling proteins, namely by discrete switching between two alternative heptad patterns, is being disclosed for the first time.

We thank Dr Buschiazzo for the very positive feed-back and the expert comments that surely helped us to improve the MS. Our point-to-point response is given below (in blue).

Major concerns

1- The authors conclude that DgcR is a constitutive dimer. This is important, because REC phosphorylation could in principle favor dimerization and thus raise the effective concentration of the two GGDEFs, ultimately increasing DGC activity. If so, this would correspond to the known 'PleD-kind' of mechanism. MW estimations to characterize DgcR's quaternary structure are thus critically important, scrutinizing whether there are monomer-dimer or dimer-tetramer equilibria. Key parameters to do so are not clear enough to me, so I do have a few questions:

what are the results in SEC-MALS when using 20mM NaCl? (20mM NaCl seems to work OK in your purification protocol). The reported 500 mM NaCl for SEC-MALS (Mat & Meth refers to 500 mL, but I assume this is a typo), is a quite high salt concentration. High ionic force will reinforce hydrophobic-mediated interactions, and potentially hold monomers together in a more stable dimeric form, even at low protein concentrations.

Yes, the salt concentration used in the previous SEC-MALS is 500 mM. This has been corrected in Materials and Methods section. We have performed another SEC-MALS analysis of DgcR (wt) and DgcR' (AxxA mutant) at very low salt concentration (20 mM NaCl) and the lowest protein concentration (0.014 mM) used previously. Clearly, the same dimeric state is observed also here (new panel c in Suppl. Fig. 2), strongly suggesting that DgcR is a constitutive dimer under physiological conditions. Also, addition of c-di-GMP did not have an effect on the oligomeric state.

please share the chromatograms corresponding to the SEC purification step (an additional Supplementary figure would be helpful)

We have included the SEC purification chromatograms to Supplementary Fig. 3. Please read our response to the related comment below.

why was a Superdex 200 column chosen, instead of an S75? I suggest repeating the separation with the latter one (at difference with S200, in an S75 the monomeric and dimeric species are expected to be included and separated with greater resolution). A potential tetramer would elute within the exclusion volume, neatly distinct from dimers and monomers.

We have compared DgcR elution done in S200 16/60 and S75 16/60. Indeed, the reviewer is right and the purification is improved when performed in a S75 16/60. Nevertheless, no tetrameric species was detected, see new Supplementary Fig. S3. A statement about the purification using S75 has been added to Materials and Methods.

how does the AxxA mutant behave in SEC-MALS? (I understand from Mat & Methods that only the wild-type protein was used) Can different quaternary structures for wild-type vs AxxA be ruled out based on direct evidence?

The requested data are now included in the new Supplementary Fig. 2c. Clearly, the same dimeric state is observed also for the mutant.

2- Precisely how are crystal packing contacts fixing in place the GGDEF domains in the native and activated AxxA structures? The authors do say that the GGDEF domains are being held in position by crystal contacts (in both structures), I believe this deserves a more elaborate description (including with an illustrative supplementary fig). This comment leads to two connected issues:

The most relevant finding of our structures is that there are no direct contacts between the domains of the protomers. We have added a sentence to this respect (line 115). Thus, communication between the domains is achieved by conformational changes in the inter-domain linker, as also stated on line 279 "The following steps invoke no direct Rec - GGDEF communication, but only an unrestricted rotation of the GGDEF domains around the inter-domain hinges. ". Influence of crystal forces on the actual orientation of the GGDEF domains appears thus not relevant for our proposed mechanism and we, therefore, don't like to diverge into such details.

(Question x) more evidence appears to be needed to substantiate the claim that the AxxA native structure represents a catalytically non-productive configuration, whereas the AxxA activated one is poised for catalysis (as an immediately previous state to the catalytically competent one; illustrated in Fig 7 in the transition from panels e to f).

This is an important question. But it seems clear that the proposed, catalytically competent arrangement of the GGDEF domains in the activated structure (Fig. 6b) should be disrupted severely and, thus, become non-productive, when changing the Rec dimer to the native structure with the concomitant substantial (9 Å) relative translation of the inter-domain hinges. We have added a respective sentence ("Noteworthy, the two GGDEF domains most likely won't be able to attain the productive arrangement when the Rec dimer is in the native form, since the inter-domain hinge is translated substantially upon activation/deactivation (Fig. 6b)." line 242) to the manuscript. See also our answer to question y.

If the GGDEF domains are indeed extremely flexible relative to the rest of the molecule, how reliable is the GGDEF orientation information, comparing the native and activated structures, to interpret steps along the activation process (fig 7)?

The crystal structures suggest that the relative movement of the GGDEF domains w/r to the dimeric Rec stem is largely restricted to a well-defined hinge rotation around psi136. This is basically the only information we are using to model the motion of the GGDEF domains towards a competent arrangement upon activation. The referee is right that the symmetrized activated structure (Fig. 7e) is a somewhat arbitrary intermediate (at least in part defined by crystal contacts) towards the competent arrangement. But this intermediate structure is not crucial for our proposed mechanism at all. An (intermediate) structure with a different hinge torsion angle would have done equally well.

could the role of REC phosphorylation in modulating DGC activity be further elaborated? This issue interrogates the section on "Structural coupling of Rec modification with competent dimer formation" (page 11-12).

It is not clear, what precise aspect of the model should be explained better. As for the effect of REC phosphorylation, we already stated "... aspartate pseudo-phosphorylation induces a rigid-body motion within each Rec domain (Fig. 7b, tertiary change)." line 273) and the subsequent consequences are discussed in detail.

(Question y) An apparently subtle difference seems to me a very important one, comparing figures 1b vs 7a. The unphosphorylated, native form of DgcR (Fig 1b and 2b,c) is actually proving that the GGDEFs can adopt a configuration poised to adopting a competent constellation, namely the one adopted by chain B (not included in Fig 7a). The authors emphasize that there are almost no direct constraints between REC and GGDEF domains, the latter enabled to move quite freely. So, at the end of the day the question seems to remain unanswered. Please clarify.

Indeed, chain B of the native DgcR structure (Fig. 1b) is in a similar, but by no means identical, conformation as each subunit of the catalytically competent model (Fig. 6b, 7f). But note, using chain B to symmetrize the dimer to have both substrates juxtaposed would result in very severe clashes. The answer to this question relates also to the answer given for question x above.

3- How come the c-di-GMP K_i figures are different for native vs active species? This appears to indicate that REC phosphorylation is engaged in modulating REC-GGDEF interaction. The authors briefly refer to suboptimal product-mediated cross-linking when the protein is not phosphorylated, but in any case, this suggests that GGDEF domains are not really independent (which contradicts the authors' statement). Please correct and/or clarify in the main text.

That the K_i is sensitive to the activation state of the Rec domains does not necessarily imply that the Rec domains interact (non-covalently) with the GGDEF domains. Rather this can be explained again by considering that the relative disposition of the inter-domain hinges is changed upon activation, which obviously will affect the structure of the c-di-GMP cross-

linked GGDEF dimer. This would be analogous to how Rec phosphorylation is believed to change the conformational space available for GTP loaded GGDEF association.

Minor comments

- The Abstract would improve by explicitly stating the knowledge gap that the authors set out to address. A phrase in this sense is expected separating the initial background sentences, and the actual results that are being reported now.

We have added a half sentence to this avail.

- Substitute reference #15 by the original source of morbidity/mortality burden : Costa et al. Global morbidity and mortality of leptospirosis: a systematic review. PLoS Negl Trop Dis 2015;9:e0003898.

We have replaced the reference.

- Following up on the previous point, ref #16 might also be substituted by a more general review wherein using *L. biflexa* as a model is also mentioned; for example, Picardeau Virulence of the zoonotic agent of leptospirosis: still terra incognita? Nat Rev Microbiol 2017 15:297.

We have replaced the reference.

- Previous work on histidine kinases is referred to by the authors as an inspiring source of information concerning mechanisms of signal transmission via coiled-coils. I candidly believe that at least two key reports from our group deserve fair credit, because they are pertinent in the context of the authors' discussion and would likely help to enrich its views and better put the work in context.

Our review (Buschiazzo & Trajtenberg Annu Rev Microbiol 2019) should probably be cited when you invoke HK phosphorylation of REC domains in the Introduction, as one of the most updated reviews in the field. Apart from alternative heptad registers, on which the authors pertinently focus on, this review will likely help in enriching the Discussion, bringing up the universal feature of non-ideal heptads (i.e. including polar residues at a/d positions, and heptad phase adjustments such as skip positions, stutters and stammers). Such departure from ideal hydrophobic packing enables iso-energetic conformational rearrangements: indeed, Fig 10a in this ms readily shows the presence of polar residues like Thr and Arg at some 'a' positions, and even more diverse variations at 'd' positions.

Indeed, this review is relevant and up-to-date. We now refer to it in the introduction (line 33).

The authors here refer to examples of coiled-coil signal-transmission in HKs, citing Diensthuber et al Structure 2013 and Wang et al PLoS Biol 2013 (line 388). However, our work on the HK DesK (Albanesi et al PNAS 2009) seems also particularly relevant, as it was the first experimental report showing the workings of signal-transmission via coiled-coil signaling helices in HKs, which had previously been predicted (Anantharaman et al Biol Direct 2006) Our work came out months after the Moffat lab presented convergent results, albeit working with chimeric LOV-HK constructs (Möglich et al 2009 J Mol Biol). Albanesi et al 2009 was followed by Saita et al Mol Microbiol 2015 and Trajtenberg et al eLife 2016, wrapping up one of the most complete examples of coiled-coil mechanistic studies in HK signaling.

Indeed, discussion of the general signalling mechanism in HKs was lacking in the MS. We thank for this comment. We have added a sentence (line 412) with reference to Albanesi and to others.

- The Methods section says that the minimum concentration of protein analyzed by SEC-MALS was 0.4 mg/mL, but figure S2 shows it was 0.5 mg/mL. Which one is correct? Moreover, Fig S2's legend refers to 0.014 mM as the lowest protein concentration, from what I deduce the authors consider the species to be a monomer (corresponding to 0.007 mM of dimer), yet showing it behaves as a dimer in SEC under these conditions.

0.5 mg/mL is correct, it has been corrected in Methods and Materials section. Indeed, the given molar concentrations refer to monomers. We have added a sentence clarifying this on Supplementary Fig. 2.

- Fig S2 caption does not describe how much BeF3- was used.

For conditions with BeF3-, sample was treated as described in Methods. This statement was added to Fig. S2 caption.

- I can only guess that crystallization of wild-type DgcR didn't work, why? considering that the AxxA variant is used as a good model of the wt, substantial structural variations between them are not expected, right?

Wild-type DgcR crystallisation was attempted only in presence of c-di-GMP (DgcR_inh). To obtain apo DgcR was found to be difficult, since the wild type co-purifies to some extent with bound c-di-GMP. Treatment with a phosphodiesterase in order to break c-di-GMP or extensive dialysis would have been required to obtain the apo protein. These additional steps are not required when working with the AxxA variant, thereby making the whole process easier. We do not expect that the mutations affect the protein structure elsewhere, in particular not the Rec, the coiled-coil linker or the active site structure.

Why was 3'dGTP added? What if it is not added?

3'dGTP was added because we were hopeful to trap the catalytically competent structure and wanted to prevent substrate turnover for this purpose. But in any case, a substrate (analog) bound structure was considered to be most relevant, considering the high intracellular concentration of GTP.

- The phosphorylation-induced reorganization in the REC domains, shifting the interaction between a4-b5-a5 faces, seems to be very similar to the one that was observed between the REC and pseudoREC domains in PleD -in which case this favored subsequent dimerization. Is this correct?

This is absolutely correct. We have added a respective discussion to the MS ("A special case is the Rec-Rec'-GGDEF protein PleD,... " lines 178), since this may be indeed interesting to more readers.

I wonder whether without the CC, DgcR's REC domains alone would otherwise undergo a P-triggered monomer-to-dimer equilibrium shift. Have you envisaged a REC-only construct of DgcR, to evaluate this? If a REC-only monomer to dimer shift were confirmed due to phosphorylation, this could be important in modifying the dynamics even if the full-length dimer is indeed constitutive (HDX-MS -such as done with IsPadC- and/or SAXS would also be confirmatory of such scenario)

We have not worked with a Rec-only construct. We don't quite see how such a truncated protein would yield dynamic information for the full-length protein, since the linker probably contributes considerably to the stability (and reduced dynamics) of the Rec dimer.

- Fig 2: the plot in panel (a) is not that clear: it would be helpful to show nat_A (to readily compare to chain B); the boxes do not correspond straightforwardly to residues on the x-axis

Indeed, the labeling x-axis was not clear. Since there are two main-chain dihedrals per residue, each residue number shows the values of two torsion angle differences (now

exemplified for residue numebr 136). We improved the legend to clearly define the y-axis (Δtor).

- Fig 3: the caption indicates Thr86, but in the text this Thr is first numerated as 85 (page 5 line 119) I guess the latter is a typo, afterwards in the text (line 137) Thr86 is mentioned.

Thr 86 is correct. Threonine number on line 119 has been corrected to T86.

A closeup into the phosphorylation site would be very useful to see the shifts (e.g. adding a panel c).

We think the description of the structural changes accompanying pseudo-phosphorylation that we give in the text and Fig. 3a showing all relevant residues and H-bonds should be sufficient.

- Fig 6 : I can't fully understand what you are plotting in panel (a): what is exactly represented with the last two boxes? Is it torsion angles for residue 137? Or yet themanually produced shifts on residue 136 phi/psi angles? The x-axis is confusing

The representation in Fig. 6a is the same as in Fig. 2a, which we have improved (see above). One can see that the manual torsion angle changes were limited to the 4 main-chain dihedrals of residues 136 and 137.

Alejandro Buschiazzo, PhD

Reviewer #3 (Remarks to the Author):

Diguanylate cyclases (DGC) catalyze the formation of the widespread bacterial second messenger c-di-GMP. DGCs, also referred to as GGDEF, act as obligate dimers with the active site formed across the dimer interface. The dimeric DGCs are found in conjunction with a palette of different N-terminal sensor/input domains to which they are connected by dimeric, parallel α -helical coiled coils.

The architecture REC-GGDEF studied presently by Schirmer and colleagues comprises an N-terminal receiver domain and is particularly common in nature. REC domains form part of two-component systems and are phosphorylated by histidine kinases to regulate the activity of associated output modules. In the present contribution, the authors report crystal structures of DgcR from *Leptospira biflexa* in its native form, a pseudo-activated form (in the presence of BeF₃⁻), and a product-inhibited form.

The detailed comparison of the structures allow the authors to infer a likely molecular trajectory via which DgcR is activated upon phosphorylation. A rearrangement of the REC dimer interface is relayed to and through the coiled coil as a change in coiled-coil register. This structural transition arrives at the GGDEF dimer and enables the adoption of the catalytically competent state. The product inhibition is apparently based on c-di-GMP-mediated inter-protomer contacts that capture the GGDEF dimer in an unproductive arrangement.

Taken together, this is an equally well conceived and executed manuscript that reports an abundance of relevant and captivating data; easily the best manuscript I have read in many weeks. The experiments have been executed to the highest standard, and likewise the manuscript is written clearly and carefully. All major conclusions are fully supported by the data. I strongly expect the work to be of profound interest to a broad readership, and I hence recommend publication in Nat Commun after suitable revision.

We thank the reviewer for the enthusiastic feed-back and the constructive comments. Our point-to-point response is given below (in blue).

Most comments I have are relatively minor and are listed below, except for two aspects:

first, the native structure of DgcR shows asymmetry of the GGDEF domains but this observation is almost brushed aside. In fact, in the mechanistic model in Fig. 7 the dimeric molecule has been symmetrized. But could it be that the observed dimer structure is not just a product of crystal packing but has biological significance? As the authors will know, asymmetry abounds among the structures of sensor histidine kinases and has been ascribed functional relevance. I encourage the authors to entertain this idea for DGCs and remark on it in the Conclusion, even should this amount to speculation.

It is correct that we entertain a symmetric native structure at the start of our mechanistic model in Fig. 7. This is for simplicity and we think that this does not compromise our reasoning. We cannot imagine any functional role for a drastically non-symmetric structure in the activation process or in catalysis, since (1) the Rec activation mechanism is independent of the GGDEF configuration and (2) a symmetric (or close to symmetric) GGDEF arrangement is required to allow juxtaposition of the two GTP substrates for the reaction to occur. We feel that there is no analogy to HK function, e.g. half-site reactivity, and would like to refrain from such a discussion.

Second, the kinetic modeling shown in Fig. 9 should be revised: in my understanding, the arrow labeled 'k_cat' should originate at SEES but then point towards P (as it currently does) and EE (rather than EES). If this assessment is correct, then the modeling should be updated accordingly.

Corrected. We thank the referee for spotting this mistake.

Minor comments:

- 34: insert 'molecules' after 'GTP'

Corrected

- 45: insert 'PAS-GAF-' before 'PHY'

Corrected

- 70: '... dimer that is activated ...'

Corrected

- 94: it would be informative to report/show the orientation of the screw axis around which the GGDEF protomers are related by 90°. Is this axis coincident with the C2 axis for the REC and coiled-coil portions?

The transformation is a 92 deg rotation with an 11 Å screw component. We don't see why the orientation of this axis is of interest. All the more, since the GGDEF domains don't contact the Rec dimers.

- 114: abbreviation MALS should be spelled out

Corrected

- 216-231: some more information on how the catalytically competent conformation was generated is warranted. Presumably this state was modelled based on a previously determined DGC structure which should be mentioned here.

Unfortunately, there is no competent DGC dimer structure. As explained, the structure was modeled by applying only small changes to the main-chain dihedrals of the hinge (see Fig. 6a) to place the two reconstructed 3'-hydroxyl groups in line with the scissile PA – O3A bond of the adjacent substrate.

- 258: see above general comment, how/why was the structure symmetrized? Is there any indication that asymmetry may play a role?

We want to refer to our answer to the general comment above. To the point, a symmetrized model was used for simplicity, since the final active dimer is by necessity symmetric.

- 265-270: why do the GGDEF domains not adopt the catalytically competent arrangement in the native state? Would a steric clash result in this arrangement?

A similar question has been asked by reviewer 2 (question x). We repeat our answer here: This is an important question. But it seems clear that the proposed, catalytically competent arrangement of the GGDEF domains in the activated structure (Fig. 6b) should be disrupted severely and, thus, become non-productive, when changing the Rec dimer to the native structure with the concomitant substantial (9 Å) relative translation of the inter-domain hinges. We have added a respective sentence ("Noteworthy, the two GGDEF domains most likely won't be able to attain the productive arrangement when the Rec dimer is in the native form, since the inter-domain hinge is translated substantially upon activation/deactivation (Fig. 6b)." line 242) to the manuscript.

- 308: can a timescale be provided for the statement 'slow transition'?

At the indicated line in the manuscript, this statement is meant purely qualitative. However, a few lines later (lines 338) reference is made to the Suppl. Table 2, which lists all parameters. In the MS, we have added the k_{off} value in brackets.

- 312-324: in the scheme in Fig. 9, it would help if some of the arrows were fused in the manner done for the arrow denoted 'kcat'. At first glance, I found the scheme quite confusing but suspect that it may be clearer if this change was accommodated. And, as noted above, it appears that the arrow 'kcat' should point to EE as opposed to EES, no?

We have changed the scheme as requested above (k_{cat} is now pointing to EE and the arrows merge).

- 321-324: what is the confidence on the parameter values?

The standard deviations for the parameters as obtained from the fitting routine have been added in Suppl. Tab. 2.

- 328: rephrase 'inhibition relieved'

We like the term "inhibition relieved". The inhibition site mutant was expected to fully relieve c-di-GMP mediated inhibition of the catalytic activity.

- 332: insert 'basal' before 'activity'

It has been corrected.

- 352: elaborate on the conserved axxdexx pattern. What is the consensus sequence for this pattern, and how does it support the alternate coiled-coil registers?

Yes, we think that the consensus sequence given in Fig. 10c fully supports the "slippery" mechanism, requiring strong hydrophobic interactions (almost exclusively Leu residues) between the permanent *a* positions and less strong interactions (that can be realized with a variety of residue types, hence the variability) between the *d/e* positions. These arguments have been added to the MS (lines 372-375).

fig. 9: it is rather surprising that at long times the non-activated wild-type DGC exhibits higher turnover than the activated one. What is the (suspected) origin of this observation?

Yes, this was unexpected. But it is fully explained by the kinetic model with the refined parameter sets. The main reason for this behaviour is that the K_i of the native enzyme is rather large (weak c-di-GMP binding), whereas it is low for the activated enzyme. So, although the k_{cat} (v_{init} !) of the activated enzyme is much larger, the non-activated enzymes wins after long incubation times, since it continues to be (weakly) active until high c-di-GMP concentration are produced.

REVIEWERS' COMMENTS

Reviewer #1 (Remarks to the Author):

Teixeira and coworkers have done a very good job in addressing my concerns and recommendations. The manuscript has improved in readability also due to their changes in nomenclature as well as the many additions and clarifications incorporated. Therefore I am confident that this will be a very interesting read for scientists interested in enzyme mechanisms, interdomain signalling and structural transitions accompanying sensor activation.

A few minor things I noticed:

line238 - old nomenclature still in use for DgcR_act

line254 - still "mutant" is used very frequently for protein variants. I would recommend to use variant, mutein or something similar but not the nucleic acid-related "mutant" term.

Supplementary figure 2 - low quality; almost non-legible. please improve for the final supplementary material

Supplementary figure 7 - the intermediate would be pppGpG!

Reviewer #2 (Remarks to the Author):

The authors have addressed most of the issues and comments, the new version is definitely clearer.

They have included new data that confirm key aspects of the protein's quaternary structure, strengthening the paper's conclusions.

I want also to highlight the sections uncovering the BeF-associated rearrangements of the Rec protomers and dimer, which are very well elaborated.

In my opinion, the phosphorylation-triggered shift in the heptad phase of the coiled-coil extension of $\alpha 5$ helices, is one of the cornerstone findings of the article.

Such shift correlates with a translational movement of the constituent helices, substantially modifying the relative tilt of the coiled-coil.

Furthermore, these observations converge charmingly with some similar mechanisms reported in sensory histidine kinases.

If I may, I only have a few minor suggestions, concerning points that could still benefit of further clarification:

1- The point I had raised before -was my second major comment- was partially addressed.

I think the paper would improve by better explaining, more straightforwardly, the molecular basis of GGDEF reorientation once Recs are phosphorylated.

This is an important element of the activation mechanism.

To put it in other words, the weakness one can still note in the current form of the ms, stems from the fact that GGDEF domains are structurally independent from the Rec domains.

Namely, in both DgcR' (line 100) and DgcR'* (line 229) the authors point out that GGDEFs are positioned in the way they are observed, because of crystal contacts, hence probably flexible to move around in solution (which seems indeed a very reasonable interpretation that the authors put forward). If both domains are "independent", how come a reorganization of one -Rec- affects the positioning of the other -GGDEF?

A focused analysis on the movement of the loop delimited by Ile137 and Asn146, and how it can be restrained by the scissors-like rearrangement of the upstream coiled-coil helices, would be very useful.

Some guiding questions in this direction: Does the coiled-coil reorganization propagate into a physical constraint of the ψ_{136} dihedral? and/or of yet other structural elements the authors might observe? Can you show in some way that once the coiled-coil adopts the Rec-activated configuration, the Ile137-Asn146 loop cannot attain conformations similar to chain A in the inactive state? etc

2- In lines 100-102 you compare ψ_{136} between chains A and B of DgcR', right? thus pinpointing the location of an inter-domain hinge.

With this purpose in mind, figure 2a -although improved- still seems a bit confusing to me: it doesn't show the A vs B comparison of the native structure.

And instead, it introduces DgcR'* overlapped onto that comparison (even though the active-state structure is analyzed later...).

Maybe three slightly smaller plots one on top of the other, would be clearer. Each one would show simple pairwise comparisons between the protomers of each dimer, one from DgcR' (A vs B), and the other two from DgcR'* (A vs B, but also chains C vs D, instrumental for the reader to appreciate the whole range of variability actually observed).

3- The caption on Supp Fig 1 should read DgcR'*

Alejandro Buschiazso, PhD

Reviewer #3 (Remarks to the Author):

The authors have well responded to the reviewer comments, thus further improving the manuscript. I recommend publication without further revision.

Reviewer #1 (Remarks to the Author):

Teixeira and coworkers have done a very good job in addressing my concerns and recommendations. The manuscript has improved in readability also due to their changes in nomenclature as well as the many additions and clarifications incorporated. Therefore I am confident that this will be a very interesting read for scientists interested in enzyme mechanisms, interdomain signalling and structural transitions accompanying sensor activation.

We thank the reviewer for the positive feedback. Our point-to-point response is given below (in blue).

A few minor things I noticed:

line238 - old nomenclature still in use for DgcR_act

We have corrected it

line254 - still "mutant" is used very frequently for protein variants. I would recommend to use variant, mutein or something similar but not the nucleic acid-related "mutant" term.

We have corrected it

Supplementary figure 2 - low quality; almost non-legible. please improve for the final supplementary material

We have corrected it

Supplementary figure 7 - the intermediate would be pppGpG!

Thank you for finding this error.

Reviewer #2 (Remarks to the Author)

We thank the reviewer for positive feed-back. Our point-to-point response to the remaining issues is given below (in blue).

The authors have addressed most of the issues and comments, the new version is definitely clearer.

They have included new data that confirm key aspects of the protein's quaternary structure, strengthening the paper's conclusions.

I want also to highlight the sections uncovering the BeF-associated rearrangements of the Rec protomers and dimer, which are very well elaborated.

In my opinion, the phosphorylation-triggered shift in the heptad phase of the coiled-coil extension of $\alpha 5$ helices, is one of the cornerstone findings of the article.

Such shift correlates with a translational movement of the constituent helices, substantially modifying the relative tilt of the coiled-coil.

Furthermore, these observations converge charmingly with some similar mechanisms reported in sensory histidine kinases.

To be clear, we actually suggest that alternative heptad interfaces are realized by a relative lateral shift, not a re-orientation, of the coiled-coil helices.

If I may, I only have a few minor suggestions, concerning points that could still benefit of further clarification:

1- The point I had raised before -was my second major comment- was partially addressed.

I think the paper would improve by better explaining, more straightforwardly, the molecular basis of GGDEF reorientation once Recs are phosphorylated.

This is an important element of the activation mechanism.

To put it in other words, the weakness one can still note in the current form of the ms, stems from the fact that GGDEF domains are structurally independent from the Rec domains.

Namely, in both DgcR' (line 100) and DgcR'* (line 229) the authors point out that GGDEFs are positioned in the way they are observed, because of crystal contacts, hence probably flexible to move around in solution (which seems indeed a very reasonable interpretation that the authors put forward). If both domains are "independent", how come a reorganization of one -Rec- affects the positioning of the other -GGDEF?

We fully agree that the raised issue is of central importance for our proposed activation mechanism. Since there are no direct contacts between the Rec and the GGDEF domains, the communication is mediated by the coiled-coil linker. With the Rec dimer in the activated conformation, the coiled-coil is in a conformation that permits the formation of a

competent GGDEF dimer (lines 231-269, Fig.7e). However, with the Rec dimer in the native conformation, this is most likely not possible (lines 286-289 ff.) meaning that the enzyme would be inactive.

To improve the flow of the MS we have now moved the sentence 242ff. to a position at the end of the 286ff. section.

A focused analysis on the movement of the loop delimited by Ile137 and Asn146, and how it can be restrained by the scissors-like rearrangement of the upstream coiled-coil helices, would be very useful.

Some guiding questions in this direction: Does the coiled-coil reorganization propagate into a physical constraint of the ψ 136 dihedral? and/or of yet other structural elements the authors might observe? Can you show in some way that once the coiled-coil adopts the Rec-activated configuration, the Ile137-Asn146 loop cannot attain conformations similar to chain A in the inactive state? etc

The conformational freedom of the REC-CC - GGDEF linker is surely affected by potential steric clashes between the two GGDEF domains (Note that the linker is rather short (137 - 138), since the rigid GGDEF domain starts already at residue 139). But we think for explaining the inactivity of the native state we don't have to resort to kinetic arguments (blockage of GGDEF movement towards the competent arrangement). Rather we point out that the relative distance between the hinges is substantially distinct in the two states (lines 286 ff) allowing only one (the activated) state to adopt a competent GGDEF dimer arrangement. I.e. that structurally a competent GGDEF dimer can not be linked to a native Rec-CC dimer.

2- In lines 100-102 you compare ψ 136 between chains A and B of DgcR', right? thus pinpointing the location of an inter-domain hinge.

Correct.

With this purpose in mind, figure 2a -although improved- still seems a bit confusing to me: it doesn't show the A vs B comparison of the native structure.

The figure shows the difference (Δ tor) between the main-chain dihedrals of the two chains, not the absolute values of the two chains. We find this most appropriate to identify the hinge.

And instead, it introduces DgcR'* overlapped onto that comparison (even though the active-state structure is analyzed later...).

To demonstrate the small changes between DgcR'(A) and the two chains of DgcR'* we found it convenient to use the same graph, instead of showing a separate figure later.

Maybe three slightly smaller plots one on top of the other, would be clearer. Each one would show simple pairwise comparisons between the protomers of each dimer, one from DgcR' (A vs B), and the other two from DgcR'* (A vs B, but also chains C vs D, instrumental for the reader to appreciate the whole range of variability actually observed).

To demonstrate changes we find difference plots more useful.

3- The caption on Supp Fig 1 should read DgcR’*

Thank you for finding this error.

Alejandro Buschiazso, PhD

Reviewer #3 (Remarks to the Author)

The authors have well responded to the reviewer comments, thus further improving the manuscript. I recommend publication without further revision.

Thank you.